# *BADI*: Black-box & Anytime-valid Dataset Identification for large language models

## Abstract

Large language models (LLMs) are trained on massive, uncurated internet datasets that often include copyrighted material, making training data identification essential for intellectual property protection. Dataset inference (DI) addresses this challenge by extracting diverse training membership features for a suspect set, aggregating them, and applying statistical tests to assess if that suspect set contributed to the model's training. However, current DI methods face two major limitations that hinder their practical deployment. First, they require gray-box access to token probabilities, while state-of-the-art LLM APIs usually return only generated tokens. We address this issue by approximating per-token probabilities from label-only outputs, making black-box DI feasible. Second, existing DIs rely on p-value for statistical tests that necessitate a fixed suspect set and a predetermined significance level. This either leads to high computational costs for large suspect sets, especially in the black-box setup, or yields inconclusive results for smaller sets, since adding new suspect data points post-hoc might be necessary to provide strong enough evidence, but it invalidates statistical guarantees based on p-values. To overcome this limitation, we introduce a black-box DI framework based on e-values and sequential testing. The e-values offer anytime-valid guarantees and support optional continuation, enabling safe accumulation of evidence, reducing inconclusive outcomes and compute costs. Through these two fundamental advances, our **B**lack-box and **A**nytime-valid **D**ataset **I**dentification (*BADI*) method enables practical data auditing for LLMs, supporting their trustworthy deployment.

## 1 Introduction

Large Language Models (LLMs) such as GPT (Achiam et al., 2023), Llama (Dubey et al., 2024; Touvron et al., 2023), Gemini (Team et al., 2024), and Claude (Anthropic, 2023) are trained on massive, uncurated datasets from the internet. These datasets often contain large amounts of copyrighted texts, raising concerns about intellectual property infringement. The risk is not only hypothetical, as recent work has demonstrated that models such as Llama 3.1 can reproduce entire copyrighted books verbatim (Cooper et al., 2025). But the problem extends beyond books: Currently, there are more than 25 active lawsuits (Sinai) on the unauthorized use of copyrighted texts, such as news articles (The New York Times v. Microsoft and OpenAI., 2023) and source code (J. Doe 1 et al. v. GitHub, Inc., Microsoft Corporation, and OpenAI, Inc., 2024; J. Doe 1 et al. v. GitHub., Microsoft, and OpenAI et al., 2024) by LLMs. These developments underscore the urgent need for technical methods that can reliably determine which data was used to train a given LLM.

LLM Dataset Inference (DI) (Maini et al., 2024) has recently emerged as a promising approach for determining whether a suspect dataset was used to train a particular LLM. DI aggregates training membership features for each data point in the suspect set and applies statistical hypothesis testing to robustly decide if the dataset contributed to model training. Despite its potential, current DI methods for generative models (Maini et al., 2024; Kowalczuk et al., 2025) face core limitations in practical settings. Notably, extracting membership features typically requires *gray-box access* to internal model details, such as full logit vectors or per-token probabilities, which is not available for most state-of-the-art LLMs that are deployed via APIs (Achiam et al., 2023; Team et al., 2024; Anthropic, 2023), where only black-box access is allowed. How to reliably extract training membership features for DI under these constraints remains an open problem. Additionally, existing DI-methods rely on *p-value-based statistical testing*, which requires specifying a fixed suspect dataset and significance

Figure 1: **Overview of *BADI*:** ❶ Sample pairs of data points from suspect and held-out sets one by one. ❷ Compute black-box membership features using tokens from the target LLM. ❸ Use scores from all previously obtained samples to train a scoring model that distinguishes between suspect and held-out data points. ❹ Bet on the scoring model's ability to distinguish suspect and held-out sets in the incoming data. ❺ Wealth accumulated in the betting process is a direct measure of evidence against the null hypothesis that the target model did not see the suspect data during training. ❻ Add more data points if a desired significance level is not met. ❼ Stop the protocol anytime (A) if we cross a desired significance threshold and declare that the suspect set was a *member set*, or (B) if we run out of suspect and held-out points or other resources. In case (B), we can still interpret the evidence, for example, as not significant and declare that the tested suspect set was a *non-member set*, i.e., not used to train the target model.

level in advance. Large suspect sets increase computational cost, especially in black-box settings where membership feature extraction is more expensive, while small sets risk inconclusive results. In principle, adding more data points could improve the testing power, but doing so invalidates the statistical guarantees associated with p-values. These challenges highlight the need for more flexible and practical DI frameworks for real-world LLM deployments.

To turn DI into a practical tool for identifying LLMs' training data, we propose our novel **B**lack-box and **A**nytime-valid **D**ataset **I**dentification (*BADI*). *BADI* enables extraction of membership features even under restricted black-box access by approximating per-token probabilities for the target LLM using only token similarities. With this approach, we are able to efficiently transform previous gray-box membership features (Carlini et al., 2021; Shi et al., 2024; Mattern et al., 2023; Mitchell et al., 2023) into their black-box counterparts. We also propose novel features for our DI setting, tailored for our black-box probability estimation. Additionally, we equip *BADI* with a novel statistical testing framework, leveraging e-values (Vovk & Wang, 2021) instead of p-values. An e-value is a nonnegative measure of evidence against a null hypothesis whose expected value does not exceed one under the null. A detailed definition is provided in section 4. E-values are particularly well-suited for our *BADI*, as they support sequential, sample-wise analyses with rigorous statistical guarantees, allowing *inference to be terminated at any point without invalidating the resulting guarantees.* Practically, this means we can begin with a small suspect set, add samples as needed, and stop testing once sufficient evidence has been obtained. This is especially an advantage in the black-box LLM setup where membership features are costly to compute. We present an overview of our method in Figure 1.

In summary, our four main contributions are as follows:

1. We propose *BADI*, a novel practical DI method for LLMs that operates with only black-box access to the models and provides rigorous, anytime-valid statistical guarantees.

2. We design novel black-box membership features and translate existing gray-box features to the black-box setup by efficiently estimating per-token probabilities from label-only outputs.

3. We equip *BADI* with a novel statistical testing framework based on e-values, which supports sequential testing, enabling flexible, sample and compute-efficient inference with anytime-valid guarantees: *allowing early stopping as soon as sufficient evidence is attained, or continuation for increased statistical evidence as resources permit.*

4. We validate our method through comprehensive empirical evaluation on LLMs with different model sizes, known held-out sets, and testing on diverse data subsets across multiple domains.

## 2 BACKGROUND AND RELATED WORK

**Membership Inference (MI).** The goal of membership inference is to detect if a given data point $x$ was used to train a model (Shokri et al., 2017; Carlini et al., 2022). Multiple MI methods have been proposed for LLMs. They mainly operate under **gray-box settings** and require access to a model's loss or perplexity. Carlini et al. (2021) leverage the observation that the ratio of a model's perplexity on a training sample to the sample's entropy (computed with the zlib library) or to its perplexity under a reference model is typically lower than for unseen texts. Shi et al. (2024) proposed MIN-K% PROB based on a simple hypothesis that an unseen text segment is likely to contain K% outlier tokens with low predicted probabilities, while an example seen during training of the model is less likely to contain such outliers. Perturbation-based MI techniques (Mattern et al., 2023) compare a model's perplexity on the original $x$ versus its perturbed version $\tilde{x}$. The perturbations may include synonym substitutions, random deletions, character-level typos, or replacements generated by an external language model (Mitchell et al., 2023). The reasoning is that training set members are more robust to such perturbations than non-members. More recently, **black-box**, label-only MI have been put forward. They usually approximate output probabilities using a surrogate model: DPDLLM fine-tunes a surrogate on the suspect text and uses text completion task (Zhou et al., 2024), while PETAL estimates the probabilities token by token on a standard GPT-2-XL model as a surrogate (He et al., 2025). However, most existing MI methods for LLMs do not outperform random guessing (Maini et al., 2024; Duan et al., 2024), including even the strong gray-box MIs. Prior studies have attributed their apparent success to distributional shifts between member and non-member data during evaluation (Maini et al., 2024; Duan et al., 2024). This significantly limits the success of MI for identifying training data in LLMs.

**Dataset Inference (DI).** To overcome the limitation of MI-based methods for training data detection, dataset inference (Maini et al., 2021; Dziedzic et al., 2022) was proposed. In generative models, it's goal is to determine whether a given suspect dataset was used to train a model (Kowalczuk et al., 2025; Maini et al., 2024). Therefore, it extracts diverse MI-features for the individual data points, aggregates them, and applies statistical testing to reliable determine whether the suspect set was used to train the model. Maini et al. (2024) were the first to successfully apply DI in LLMs. By combining 52 MI features, they were able to identify individual data subsets as training data of a given LLM. Yet, similar to DI approaches from other domains (Kowalczuk et al., 2025), their method faces severe limitations in practical applicability. First, it relies on gray-box MI features whereas state-of-the-art LLMs like GPT, Gemini, and Claude are deployed behind APIs that provide only black-box access. Second, they rely on statistical testing with p-values which is limited to a fixed size of the suspect set and predefined significance level. This causes problems because it is not possible to add more data points to the suspect set when the initial test fails without invalidating the statistical guarantees We overcome these limitations and make DI practical by extracting black-box features and performing e-value based statistical testing which provides *anytime-valid* guarantees.

**E-values.** An *e-value* (Vovk & Wang, 2021) is a nonnegative random variable $E$ such that for a given null hypothesis $H_0$ with a set of distributions $\mathcal{P}$, every distribution $P \in \mathcal{P}$ follows $H_0 : \forall P \in \mathcal{P} : \boldsymbol{E}_P[E] \leq 1$. Larger $E$ provide evidence against the null hypothesis, and a level-$\alpha$ test rejects $H_0$ if $E \geq 1/\alpha$. Consider randomly ordered data points $Z_1, \ldots, Z_n \sim P$. At step $i \in [n]$, we compute an e-value $E_i$ with respect to the information in $Z_1, \ldots, Z_i$. An *e-process* is a sequence $E_{t \geq 1}$ with the property that for *any stopping time* $\tau$, $E_\tau$ remains a *valid* e-value, i.e., $\forall P \in \mathcal{P} : \boldsymbol{E}_P[E_\tau] \leq 1$. This *optional-stopping* property of e-processes lets us test the hypothesis sequentially without fixing the number of suspect data points $n$ in advance. We may stop and reject the null hypothesis as soon as $E_t \geq 1/\alpha$, or continue if we can collect additional suspect data points and enough resources are available. Moreover, the realized e-value holds evidence regardless of any predefined threshold $1/\alpha$. So we can identify the significance level $\alpha$ post-hoc, and adjust it based on the data (Koning, 2024). A very natural way of designing sequential tests with e-values is by setting up a betting game and associating statistical evidence with wealth accumulated in the game. This introduces an intuitive

interpretation of an e-process as a sequence of bets, i.e., *testing by betting* (Shafer, 2021). *BADI* relies on this approach to obtain the benefits of e-values while keeping the simple intuition behind the framework. In the next sections, we first highlight our design of black-box MI features for *BADI* and then provide thorough details on our e-value-based statistical testing framework.

## 3 BLACK-BOX MEMBERSHIP FEATURE EXTRACTION

While pure MI methods were shown unsuccessful to reliably detect whether a data point was used during training of an LLM (Maini et al., 2024), in aggregation, they still can provide reliable DI-features. To enable black-box DI, we design a series of black-box membership inference features, including *sequence-level features*, and *token-level features*, which we aggregate in a scoring model to provide strong evidence on the presence or absence of a suspect dataset in the training data of a given LLM.

**Sequence-level Black-box Features.** We evaluate the LLM's ability to complete a given sequence by measuring the similarity between its generated continuation and the true suffix. In the *standard* approach, we use this similarity as an indicator of membership. In the *sequence-perturb* setting, we additionally assess how this similarity changes when the input prefix is slightly altered. Member sequences generally produce higher similarity scores and demonstrate greater sensitivity to such perturbations compared to non-members.

**Token-level Black-box Features.** We estimate per-token membership signals without relying on explicit access to token probabilities from the target LLM. Specifically, we adapt a range of gray-box membership inference features to the black-box setting by substituting black-box-derived probability estimates for those typically obtained in gray-box scenarios. For example, the MIN-K% PROB (Shi et al., 2024) membership inference, which leverages log likelihood of the $K\%$ least-likely tokens in the text, where their high average suggests that the text was likely included in the pretraining data. Notably, we introduce a new feature ***STRIP-K% PROB*** tailored for our black-box probability estimation. Our new feature reduces token outlier influence by discarding the most extreme $K\%$ token probabilities (both tails), yielding a stronger signal than MIN-K% PROB. We provide further details on all of the per-token black-box features that we use in the Appx. A.2, including perturbation based approaches (Mattern et al., 2023; Mitchell et al., 2023), reference-model and zlib-compression-based features Carlini et al. (2021).

**Estimating Token Probabilities.** To estimate token probabilities under black-box access, PETAL (He et al., 2025) leverages a surrogate model $M_s$ that provides per-token probabilities. Given a prefix $p = x_{t-1}, \ldots, x_1$ and ground-truth next token $x_t$, $M_s$ generates $\hat{x}_t$. PETAL then computes the semantic similarity $sim(x_t, \hat{x}_t)$ between the surrogate model's response and the true token. These similarity scores are monotonically mapped to token probabilities $P(\hat{x}_t)$ via a learned function $f$, which is then applied to the target model $M_t$. The original PETAL method utilizes linear regression to construct the mapping function $f$. For $M_t$, given $p$, it generates the next token $\hat{x}'_t$ and its estimated probability is $\tilde{P}(\hat{x}'_t) = f(sim(x_t, \hat{x}'_t))$. The main limitation of *PETAL* is that it relies on the surrogate model $M_s$ to create the mapping function $f$. This introduces significant compute overhead and, as we demonstrate in our empirical evaluation in Appx. A.6, is less reliable for many datasets. Instead, we introduce a novel ***sigmoid-based calibration*** method that calibrates token probability estimates without relying on a surrogate model. Specifically, we obtain per-token embeddings using a sentence embedding encoder. For each next token, we compute the semantic similarity (using the dot product) between the embedding of the predicted next token from the target model and that of the corresponding ground truth token. This similarity score is then transformed into a probability using the following sigmoid-based function:

$$\tilde{P}(\hat{x}'_t) = \sigma\left(2sim(x_t, \hat{x}'_t)\right) = \frac{e^{2sim(x_t, \hat{x}'_t)}}{1 + e^{2sim(x_t, \hat{x}'_t)}}, \tag{1}$$

where $\sigma(x) = \frac{e^x}{1+e^x}$ denotes the sigmoid function. Note that we directly use the ground-truth next token $x_t$ and not a token $\hat{x}_t$ predicted by a surrogate model $M_s$, eliminating the usage of $M_s$ and enormously simplifying and accelerating the approximation of token probabilities compared to PETAL.

**Scoring Model for Feature Aggregation.** To effectively aggregate the black-box features, we compute the weight of each feature according to its importance and calculate a final membership

score based on the weights. Following the gray-box LLM DI (Maini et al., 2024), we obtain the weights for black-box features via a scoring model. Unlike the original gray-box DI approach that employs a static train-test split for the scoring model, we introduce a novel ***online scoring framework*** that processes data points sequentially as they come or even request more if necessary. Specifically, we process the combined suspect and held-out sets sequentially, where for each pair of data points $Z_i$, we: (1) train a linear regressor as the scoring model on all previous sample pairs $\{Z_1, Z_2, \ldots, Z_{i-1}\}$ to learn feature importance weights, where suspect samples are labeled 0 and validation samples 1, (2) apply the trained scoring model to predict the membership score for the current $Z_i$, and (3) update our scoring model with the newly observed $Z_i$ for subsequent predictions.

This online formulation addresses the major limitation of prior work that all data points are available upfront for training the scoring model and performing the tests. Our approach is particularly advantageous for practical scenarios where we stop when enough evidence is gather, data arrives incrementally, and decisions are made according to the newly updated knowledge base. We further pre-process the black-box features to provide a more reliable input for the scoring model with two steps: (1) We first normalize all feature values such that different features are at comparable scales, and (2) we replace outliers from top and bottom of the distribution with feature means to prevent skewed correlations. After obtaining the weight for each feature through this online procedure, we produce one aggregated membership score for each data point. Those scores form the input for the statistical testing described in the next section.

# 4 ANYTIME-VALID STATISTICAL TESTING VIA E-VALUES

In DI, the problem is to decide whether the membership features, obtained from the scoring model, of the suspect dataset $X = \{X_t\}$ differ distributionally from those of a held-out dataset $Y = \{Y_t\}$. Classical DI methods use fixed-sample tests (often a t-test) and report p-values (Maini et al., 2021; Dziedzic et al., 2022; Maini et al., 2024; Kowalczuk et al., 2025; Dubiński et al., 2025). These approaches suffer from two fundamental drawbacks: (1) they require fixing the suspect set size and significance level *in advance*, and (2) they do not support "*peeking*," i.e., monitoring the evidence from the statistical test sequentially without inflating type-I error or invalidating the guarantees. This severely limits practicality, especially in black-box settings, where the cost of computing membership features is very high, making it prohibitive to use large suspect sets as in the canonical DI.

To address these limitations, we propose an anytime-valid statistical framework based on sequential two-sample testing (Shekhar & Ramdas, 2023). Evidence is collected incrementally from the samples' membership scores, and testing is stopped as soon as sufficient evidence is gathered. The evidence in our framework is measured via e-values (Vovk & Wang, 2021), which support optional stopping and maintain valid type-I error (false-positive) control even with post-hoc significance claims. Our approach does not require to specify the size of the suspect set or the significance level up front, providing both flexibility and rigorous statistical guarantees. While we introduce this statistical framework alongside our black-box LLM DI features, the framework is universal and can be applied to enhance prior DI approaches across domains and modalities. In the following, we detail the main components of our framework and describe the more technical details in Appx. C.

**Testing by Betting.** We adopt an *anytime-valid* testing framework based on recent advances on *e-values* (Vovk & Wang, 2021; Shafer, 2021; Shekhar & Ramdas, 2023). An *e-value* is a nonnegative statistic whose expectation under the null hypothesis is at most one. Intuitively, it quantifies evidence against the null: large e-values suggest the null is implausible, while small values are consistent with it. Following the betting framework (Shafer, 2021; Shekhar & Ramdas, 2023), hypothesis testing can be viewed as a *betting game* against the null. At each round $t$, the arbitrator observes a pair of data points $Z_t = (X_t, Y_t)$ and selects a payoff function $S_t$ with null-conditional expectation at most one. The bettor's wealth is updated multiplicatively as

$$W_t = W_{t-1} S_t(Z_t), \quad W_0 = 1. \tag{2}$$

The resulting sequence $\{W_t\}_{t \geq 0}$ is a nonnegative martingale under $H_0$, known as an *e-process* (Ramdas et al., 2020; Ramdas & Wang, 2025).

**Betting Strategies.** A good betting strategy is a one that ensures quick growth of evidence under the alternative. This can be quantified by log-optimal principles (Kelly, 1956) and their modern developments (Shafer, 2021; Waudby-Smith & Ramdas, 2024; Grünwald et al., 2024). Motivated

by those principles, we focus on strategies that approximately maximize expected log-wealth. Such strategies are conservative in the sense that they never stake all capital and therefore avoid ruin (zero wealth). The resulting wealth process $W_t^\star$ is a (test) martingale under $H_0$ and grows at an exponential rate under fixed alternatives. We adopt the stopping time

$$\tau := \inf\{t \geq 1 : W_t \geq 1/\alpha\},$$

which, by Ville's inequality (Ville, 1939), induces a level-$\alpha$ sequential test.

**Instantiation for DI.** In our adaptation of the general two-sample betting framework of Shekhar & Ramdas (2023) to DI, we use a linearized payoff of the form

$$S_t(Z_t) = 1 + \lambda_t u_t,$$

where $u_t$ is a bounded score quantifying the difference between $X_t$ and $Y_t$ (given by a kernel-MMD witness (Gretton et al., 2012)). Intuitively, one can think of $u_t$ as the outcome of the betting game that the arbitrator is betting for. In our case $u_t \in [-1, +1]$, where $u_t$ is close to 1 if the arbitrator correctly distinguishes member and non-member points from the pair $Z_t$, otherwise $u_t$ is close to $-1$. The value $\lambda_t$ is the stake, i.e., how much wealth the arbitrator is willing to bet, chosen adaptively from past data. This part is especially important as, the more outcomes of bets the arbitrator observes, the better they understand the data distribution, and the better of a strategy they can choose. We use an Online Newton Step (ONS) (Hazan et al., 2007) for staking, ensuring conservative but effective wealth allocation. When the suspect and held-out distributions differ, the expected log-wealth grows linearly with $t$, so wealth increases exponentially and the test is sample-efficient. Under the null, the wealth remains close to 1, preserving Type-I control (of false positives). Please, see Appx. C.1 for more details.

**Anytime Validity and Post Hoc Interpretability.** Our statistical testing framework has two main advantages over fixed-sample p-value tests: *(1) Anytime validity* means that we can monitor evidence sequentially, stop early if sufficient evidence accumulates, or continue as long as resources allow, without undermining statistical guarantees. This results from the fact that the wealth process is a nonnegative martingale under $H_0$, and as stated above, by Ville's inequality (Ville, 1939), we can stop at any time $\tau$ and reject the null if $W_\tau \geq 1/\alpha$ while still controlling the false positive rate at level $\alpha$. Furthermore, our framework offers *(2) Post Hoc Interpretability* is useful if we do not want to predefine a significance level, or have to stop testing because we run out of resources. Then, we can still report the e-value as valid evidence. This can also allow for higher significance claims. For example, imagine running a test until obtaining $e = 500$. In this case, we can report a significance of $0.002$. In contrast, if we had done a p-valued test, with prespecified threshold of $0.05$, but obtained a much lower p-value $p = 0.002$, we could only report the significance level of $0.05$, even though it intuitively looks like we have much stronger evidence. Together, these properties make our e-value-based statistical framework a natural fit for black-box DI, where data efficiency and rigorous error control are equally critical.

## 5 Empirical Evaluation

We demonstrate that our proposed *BADI* method reliably determines whether a given suspect dataset was used in the training of an LLM. We further compare our approach to other methods, showing that it consistently outperforms them. Additionally, we provide an ablation study of our *BADI* across different model sizes. We begin by presenting our experimental setup.

### 5.1 Experimental Setup

**Models and Datasets.** We evaluate our method on the Pythia models (Biderman et al., 2023), trained on subsets of the Pile dataset (Gao et al., 2020). The Pile provides the training and validation splits for Pythia, which we use to instantiate the suspect and held-out sets. The Pile dataset covers a large spectrum of domains, including academia, medicine, legislation, and code, among others.

**Baselines.** We evaluate our *BADI* against two other methods for black-box LLM training data detection. Each of the other methods designs a specific feature to measure the membership signal: (1) **Baseline** calculates the similarities between the ground truth sequences and LLM continuations using RoBERTa scores (it does not use the probability estimations), and (2) **PETAL** estimates the

probabilities with next-token semantic similarities and calibrates them with a surrogate model (it only uses a single feature and does not need a scoring model). Our *BADI* uses the sigmoid-based calibration and online scoring to aggregate our 31 black-box features. We also provide more details in Appx. B, including additional results for the **CatShift** method, which computes the similarities of LLM continuations before and after finetuning on the suspect set. However, CatShift occurred to be largely ineffective and we report its results only in Appx. B.3, focusing here in the main paper on Baseline and PETAL.

**Scoring Model and Two-sample Testing by Betting.** We adopt an online training protocol for the scoring model and our anytime-valid sequential testing. We present further technical details on the scoring model in Appx. A.4. At round $t$ of the DI, we first observe a pair of data points $Z_t = (X_t, Y_t)$ that we evaluate the current model on to compute the payoff and update the wealth. Then, we finetune the scoring model on all the available data points for 30 epochs. This procedure is repeated until we pass a desired significance level (e.g., $\alpha = 0.05$ or $\alpha = 0.01$). We estimate the statistical power using Monte Carlo simulations with $T = 1000$ independent trials. In each trial, a length $N_{\max}$ of data stream is formed via a random subset without replacement, the sequential test is run with the fixed threshold $1/\alpha$, and the stopping time is recorded if the threshold is crossed; otherwise the trial is counted as a non-rejection at $N_{\max}$. In all experiments, we use kernel MMD with a degree 2 polynomial kernel as the payoff. We map the raw payoff to $[-1, 1]$ via a $\tanh$ squashing, and employ the ONS (Online Newton Step) as the betting strategy with a stake constraint $\lambda_t \in [-\lambda_{\max}, \lambda_{\max}]$, where $\lambda_{\max} = 0.8$. Larger $\lambda_{\max}$ corresponds to more aggressive staking when a signal is present. To evaluate robustness with respect to data order, all experiments are repeated with 50 different random seeds. We report the average wealth (computed in log-space across seeds) together with log-space confidence intervals.

## 5.2 EFFECTIVENESS OF TESTING BY BETTING

The main advantage of using e-values for hypothesis testing lies in their *anytime validity*: we may continue collecting data indefinitely without inflating the type-I error rate (false positives) and reject the null hypothesis as soon as the e-value crosses the significance threshold (e.g., $W_t \geq 1/\alpha$, so $W_t \geq 100$ corresponds to $\alpha = 0.01$). As discussed in Appx. C, when an effect exists, the wealth (or an e-value) grows exponentially, whereas under the null hypothesis $H_0$ it does not systematically increase and remains near its initial value ($\leq 1$).

This behavior is illustrated in our main result in Figure 2. Across 12 subsets of the PILE dataset (with additional results in Appx. D), we observe a clear effect: the wealth for comparing membership features of members versus non-members grows exponentially. Conversely, in a negative-control setting (non-members vs. non-members), the wealth stays close to one, in line with the theoretical guarantees of sequential testing. Consequently, the observed empirical false positive rate is very low; as we present in Table 4, the sequential test achieves a false positive rate of approximately $1\%$ at the $5\%$ significance level.

In contrast to prior work on DI, which required at least 1000 samples for verification (Maini et al., 2024; Zhao et al., 2025), we show that much fewer samples are required for individual datasets in our case and we do not have to decide on this number upfront. For example, for significance level of $5\%$, this numbers are approximately $175 \pm 68$ for PhilPapers, $305 \pm 70$ for Stack Exchange, $92 \pm 21$ for Ubuntu IRC, and $461 \pm 126$ for Wikipedia. This adaptability of our method is particularly valuable in scenarios where data is scarce or where computing membership features is computationally expensive, such as in the black-box setting. For the ArXiv, Pile-CC, OpenSubtitles, FreeLaw, and GitHub datasets, we reach the $1\%$ significance threshold less frequently within the available 2000 examples. This indicates the need of incorporating additional data points to strengthen the analysis. This is the task that our method is well-suited to address, as it enables seamless accumulation of further evidence as needed.

## 5.3 EFFECTIVENESS OF OUR BLACK-BOX FEATURES

We compare the effectiveness of our black-box features and sigmoid-based probability estimation against other approaches in Figure 3. Note that PETAL uses only the probability estimation with the surrogate model and linear regression, while our method leverages many black-box features aggregated with the scoring model. Additionally, Baseline leverages only the *standard* Sequence-level

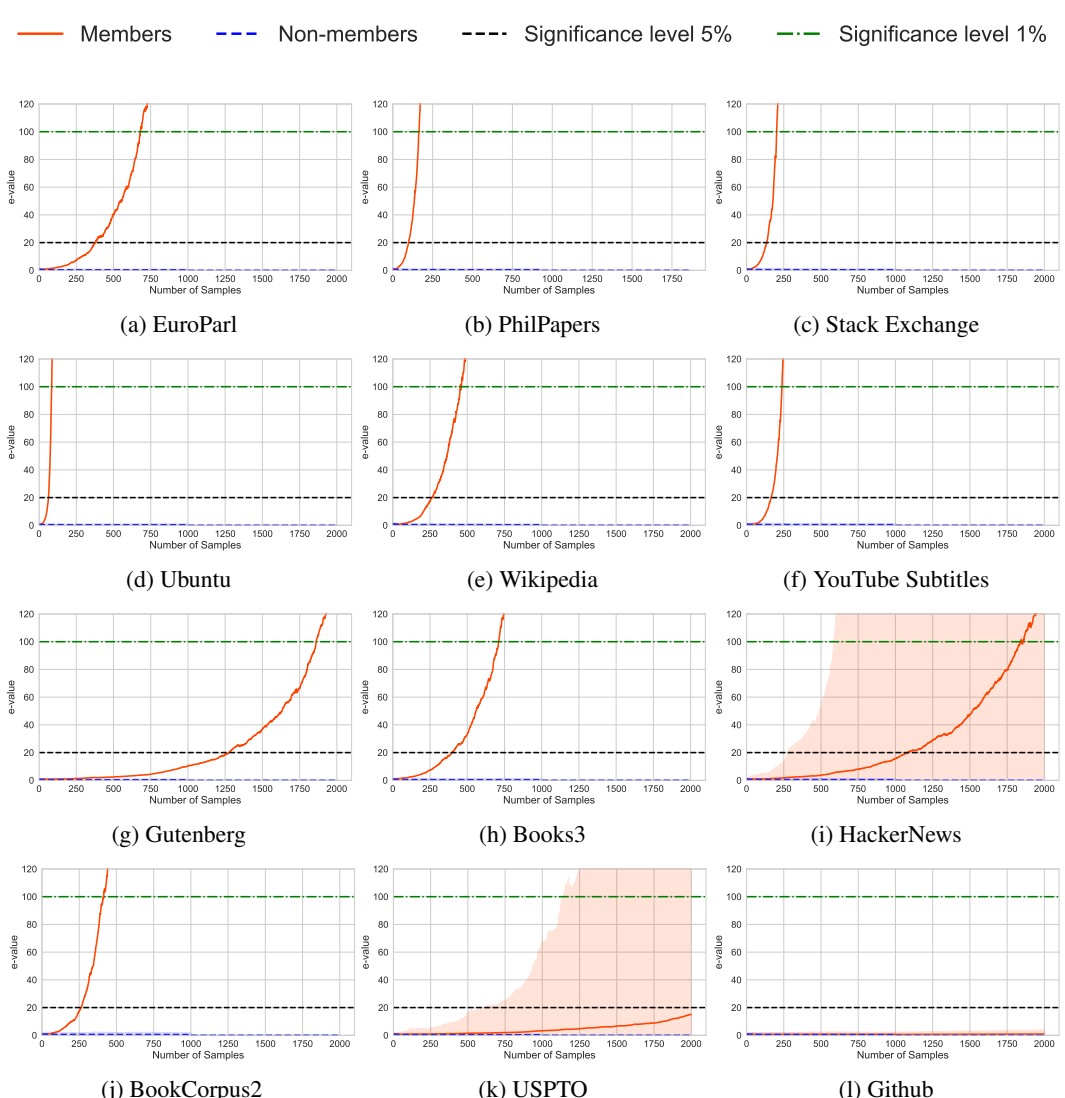

Figure 2: **Our Testing by Betting.** We present accumulated wealth with 95% confidence intervals (shown as the orange area in the background of the graphs) for testing by betting across different data subsets from the Pile for the Pythia-12B model.

black-box feature, which is used in our method along with *sequence-perturb* features. Our results highlight that given enough data points, our method provides strong enough evidence and successfully predicts the membership for data subsets from the Pile, demonstrating effectiveness of *BADI* on diverse types of texts. Notably, our method successfully prevents false positives in all cases, which means that we do not falsely accuse the model owner of using copyrighted materials. Compared with PETAL and Baseline on the Pythia-12B model, our approach reaches a given significance level with substantially fewer data points, while requiring much less compute than PETAL (we do not use the surrogate model) and adding a relatively negligible overhead with the scoring model to aggregate many more features than both PETAL and Baseline. For example, on Ubuntu IRC, the average stopping time is 92.33 observations, much fewer than 1000 required for the original LLM DI. As another example, on Books3 the Baseline shows no significant effect within 2000 observations, whereas our method achieves TPR=84% at FPR=1% with an average stopping time of only 788.47 observations.

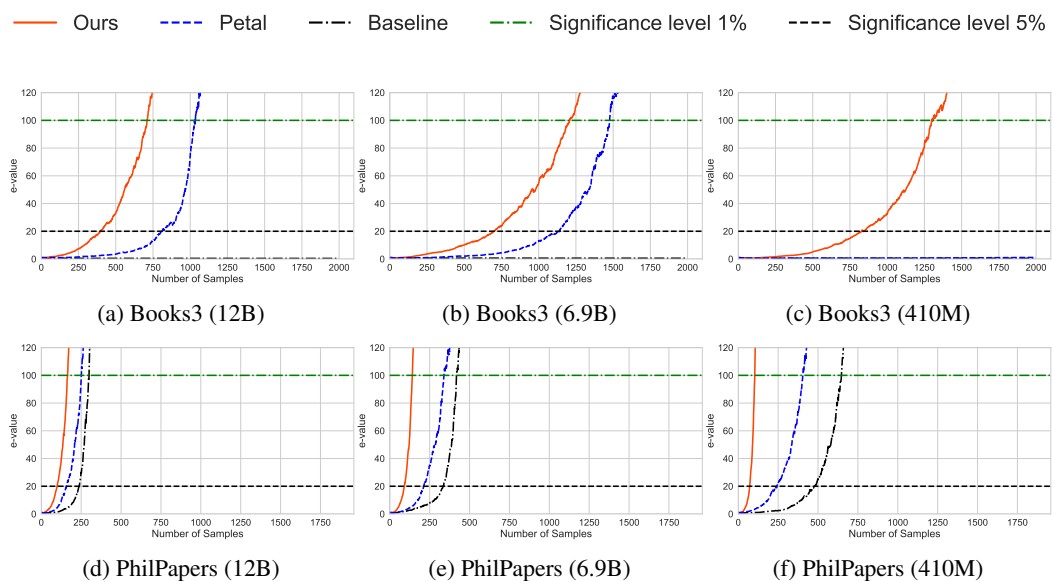

Figure 3: **Comparison Across Methods and Model Sizes.** We compare our method against PETAL and Baseline, presenting accumulated wealth for Pythia-12B (first column), Pythia-6.9B (second column), and Pythia-410M (third column) models tested on Books3 (first row) and PhilPapers (second row) datasets.

## 5.4 GENERALIZATION TO DIFFERENT MODEL SIZES

We further validate our detection method on three variants of the Pythia models, namely Pythia-410M, Pythia-6.9B, and Pythia-12B. We assess how model scale affects the performance of our method. As shown in Figure 3, our method consistently provides correct dataset identification across different models sizes. We report additional results for more datasets in Appx. D.2. Overall, scaling up the model sizes improves detection on Books3, BookCorpus2, and Gutenberg datasets, with shorter average stopping times. Detection is consistently strong on Ubuntu IRC, StackExchange, PhilPapers, and YouTube Subtitles, with early stopping. However, the scale effects are not uniform: for Wikipedia (and in some cases EuroParl and USPTO), larger models yield a weaker signal.

## 5.5 GENERALIZATION TO DIFFERENT MODEL FAMILIES AND ACCESS LEVELS

We evaluate BADI across distinct model families and access levels to assess its ability to generalize beyond a single architecture. In particular, we report detailed results on the OLMo-7B model (with white-box access) and OpenAI's GPT-3.5-Turbo API (with only the black-box access) in Appx. E. These experiments highlight BADI's applicability not only to locally deployed open-weight models, but also to real-world commercial black-box services where internal parameters and logits are inaccessible.

## 5.6 ABLATION STUDIES

**Feature Importance Analysis.** We investigate which black-box features are most significant by training the scoring model on 1000 examples across 20 random data shuffles. We then extract the average weight from these shuffles for each dataset. The feature importance weights are shown in Figure 9 for member vs. held-out data and in Figure 10 for non-member vs. held-out data (in Appx. A.5). We observe that all of our 31 black-box features are necessary, including our newly introduced STRIP-K% PROB, and all of the features provide a significant signal for different datasets.

**Token Probability Estimation.** Additionally, we compare three methods for estimating per-token probabilities. Each of them assigns probabilities to the next token $\hat{x}_t$ from the target model based on the ground truth next token $x_t$ and different transformations of token similarities. The methods are as follows: (1) the raw approach: $1 - sim(x_t, \hat{x}_t)$; (2) the PETAL-based approach: $-sim(x_t, \hat{x}_t) \cdot \beta - \alpha$,

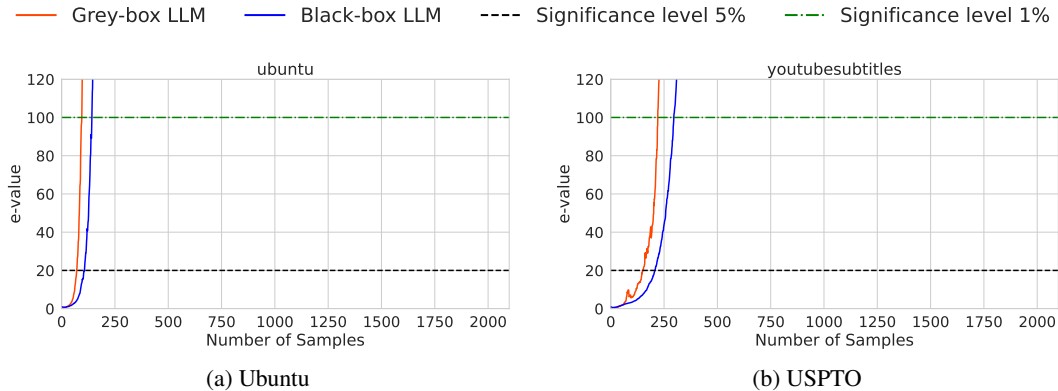

(a) Ubuntu                                    (b) USPTO

Figure 4: **Gray-box vs Black-box.** Comparison of wealth accumulation under gray-box and black-box access for the Pythia-12B model. The gray-box setting reaches the significance threshold more rapidly, yet the gap from the black-box setting remains small, highlighting the effectiveness of our method in estimating per-token probabilities even without internal model access.

where $\beta$ and $\alpha$ are parameters fitted via linear regression using surrogate probabilities; and (3) our sigmoid-based approach: $-\sigma(2, sim(x_t, \hat{x}_t))$, based on Equation (1), where $\sigma$ denotes the sigmoid function. We show in Figure 7 (in Appx. A.6) that our sigmoid-based method achieves the highest performance among the three tested approaches.

**Comparison with Gray-Box DI.** We compare BADI's evidence accumulation in a *gray-box* setting, where ground-truth token probabilities are accessible, against our *black-box* setting, where only sampled outputs are available. As shown in Figure 4, the gray-box variant accumulates wealth more rapidly, achieving statistical significance with fewer samples. Nevertheless, the performance gap remains small, demonstrating that BADI can accurately approximate per-token probabilities from black-box queries with limited estimation error. This result highlights the practicality of our approach in real-world scenarios where internal model logits are not exposed.

## 6 CONCLUSIONS AND OUTLOOK

We have introduced *BADI*, a novel black-box, anytime-valid DI framework for detecting whether a specific dataset was used to train an API-based LLM. Our key innovation is the integration of black-box membership features with sequential and anytime-valid inference, enabling us to test any black-box LLM using a flexible betting protocol that delivers valid results after any number of evaluated samples. This approach significantly reduces the computational burden of black-box inference by allowing early stopping as soon as sufficient evidence is attained, or continuation for increased statistical evidence as resources permit. Our empirical evaluation on the Pythia model suite, trained on diverse datasets from the Pile, confirms the effectiveness and scalability of our approach. As LLMs become ever larger and more opaque, white-box or gray-box DI methods are increasingly impractical. *BADI* provides a practical black-box solution for real-world auditing and compliance, supporting individuals' rights to control how their data is used in training of LLMs.

## 7 REPRODUCIBILITY STATEMENT

To ensure full reproducibility of our results, we provide comprehensive implementation details and complete source code. All experimental procedures, including the black-box feature extraction methods, sigmoid-based probability estimation, and statistical testing protocols, are described in detail in the Appx. A. Additionally, we submit the complete source code as supplementary materials, which includes all scripts for data processing, feature extraction and evaluation pipelines used to generate the results presented in this paper, allowing direct reproduction of all experiments.

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

## A    IMPLEMENTATION DETAILS

### A.1    DATASET

We use a dataset from PILE, downloaded from `https://huggingface.co/datasets/pratyushmaini/llm_dataset_inference/resolve/main`, which is the same dataset as used in the LLM Dataset Inference (Maini et al., 2024). Each subset has ∼2000 member samples and ∼2000 held-out samples, and each sample has ∼512 tokens. For the features based on similarity, we set the maximum length to 512 tokens. For the features based on BERT score, we set the suffix to 64 tokens. For this reason, similarly to previous work, we do not include Enron and NIH data subsets in our evaluation. All samples in the NIH data subset have fewer than 64 tokens and Enron has fewer than 1000 samples.

### A.2    PER-TOKEN BLACK-BOX MEMBERSHIP FEATURES

We use the following per-token black-box features:

1. **Perplexity.** A basic feature that computes the perplexity or the average loss on the target model.
2. **MIN-$K$% PROB.** (Shi et al., 2024). Averages log likelihood of the $K$% least-likely tokens in the text. A high average suggests that the text was likely included in the pretraining data. In our black-box setting, we find that $K = 0.6$ works best.
3. **STRIP-$K$% PROB.** *This is our new added feature for the black-box setting.* It reduces outlier influence by discarding the most extreme $K$% token probabilities (e.g., both tails) before aggregation (e.g., via averaging), yielding a stronger signal than MIN-$K$% PROB.
4. **Perturbation-based.** (Mattern et al., 2023; Mitchell et al., 2023) Measures the change in perplexity between the original text and its perturbed variant (e.g., via synonym substitution, random deletion, or paraphrasing by another language model).
5. **Reference-model-based.** Compares the suspect model's perplexity to that of a reference model on the given samples.
6. **Zlib ratio.** (Carlini et al., 2021) Computes the ratio between the model's perplexity and the entropy (number of bits bits) of the text after `zlib` compression (Gailly & Adler, 2004). The underlying intuition is that a model trained on particular data encodes information about it in a more efficient manner than generic compression algorithms such as `zlib`.

### A.3    TEXT PERTURBATIONS

For every prefix, six variants of perturbed text are created using NL-Augmenter `https://github.com/GEM-benchmark/NL-Augmenter`. The perturbations include:

1. **Synonym Substitution.** Words in the text are replaced with their synonyms while keeping the overall meaning intact. In our work we change words with probability 0.25.
2. **Butter Fingers.** Characters are randomly swapped, replaced with neighboring characters to simulate human typing errors. For example for character 'g' it stays unchanged or is swapped with given probability (we use 0.1) with one of following: t, b, f, h, e, d, c, y, j, n.
3. **Random Deletion.** Certain words or characters are randomly removed from the text. This introduces incomplete information to check if the model can still infer context. We delete randomly with probability 0.25.

4. **Change Character Sase.** Letters are randomly capitalized or lowercased (e.g.,'which' → 'WhIch'). Every character can be permuted with predefined probability (we use 0.1).

5. **Whitespace Perturbation.** Extra spaces are inserted or existing ones are removed. This disturbs token boundaries and formatting without altering the words themselves. (e.g. 'because the problem' → 'be ca use thep roblem'). There are separate probabilities for whitespace remove and addition. We use 0.1 and 0.05 respectively.

6. **Underscore Trick.** Spaces are randomly replaced with underscores (e.g. 'Hello world' → 'Hello_world'). Each replacement happens with given probability (we use 0.25).

### A.4 Our Scoring Model

The scoring model in our experiments is a linear layer that maps membership features of a each data point from a suspect/held-out pair to a scalar score. The sequential test compares the mean scores of suspect versus held-out via applying kernel MMD to the combined membership score. In the online training setting, the model is first evaluated, the sequential test is performed, and then the model is fine-tuned on the observed data points. Since the stakes are chosen adaptively starting from 0, we can start with a random layer, without any pretraining. Training uses binary cross-entropy loss with the Adam optimizer (Kingma & Ba, 2014), and each data point is fine-tuned for 30 epochs and a learning rate of 0.01. This process continues until the significance threshold is reached. Although the procedure could terminate at this point, we extend training and testing beyond the threshold for demonstration purposes.

### A.5 Feature Importance Analysis

The black-box feature importance weights are shown in Figure 9 for member vs. held-out data and in Figure 10 for non-member vs. held-out data.

### A.6 Methods for Token Probability Estimation

To test all gray-box features, we utilize per-token similarity to emulate the loss. First, for a given prefix $x_1, \ldots, x_{t-1}$ only the next token $\hat{x}_t$ is predicted with "greedy" decoding. To get similarities we create embeddings of the output text and oracle using the embedding model `sentences-transformers/all-MiniLM-L6-v2` and then calculate the cosine similarity per each token. We noticed that similarity differs significantly from log-probability, as shown in Figure 5. Similarity has many values of 1.0 when the model predicts the correct token, approaching ∼50% in different datasets.

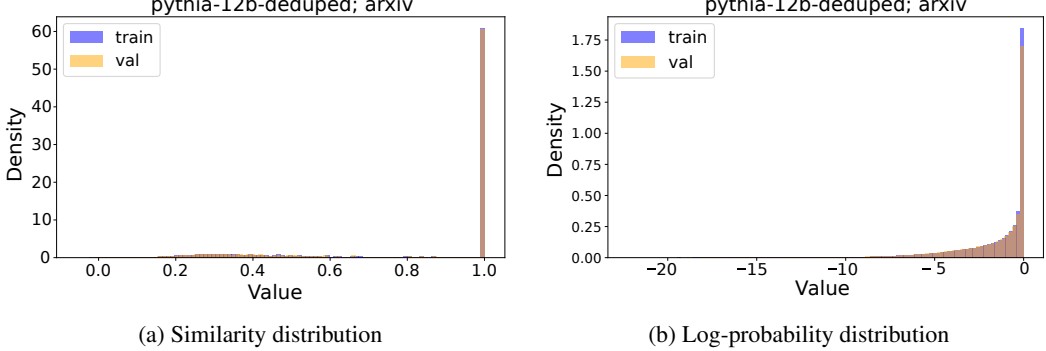

(a) Similarity distribution      (b) Log-probability distribution

Figure 5: **Comparison of Similarity and Log-probability Distributions.** We use the Pythia-12B-deduped model on arXiv dataset for this experiment.

To address this issue, three distinct methods of estimating the loss function with tokens similarity were tested:

**Raw Similarities.** As tokens are more semantically similar, a cosine similarity goes to 1 and log-loss goes to 0, therefore, the simplest approach was to transform similarity in the following way:

$$-\tilde{p}(\hat{x}_t|x_1,\ldots,x_{t-1}) = 1 - sim(x_t, \hat{x}_t), \tag{3}$$

where:

- $\tilde{p}$ - estimated probability,

- $x_1, \ldots, x_{t-1}$ - prefix,

- $\hat{x}_t$ - predicted next token,

- $x_t$ - ground truth next token,

- $sim$ - similarity function (cosine similarity).

**Reference Model.** This approach follows the original PETAL method. Per-token similarities and oracle token probabilities from the reference model (GPT2-XL) are obtained. Then linear regression is fitted to obtain slope ($\beta$) and intercept ($\alpha$). Finally log loss is estimated in following way:

$$\tilde{p}(\hat{x}_t|x_1,\ldots,x_{t-1}) = sim(x_t, \hat{x}_t) \cdot \beta - \alpha \tag{4}$$

**Sigmoid.** Sigmoid function ($\sigma(x)$) takes any real-valued number and maps to range $(0, 1)$. That property makes it an excellent choice to model the probability, then the loss estimation is as follows:

$$\tilde{p}(\hat{x}_t|x_1,\ldots,x_{t-1}) = \sigma(2sim(x_t, \hat{x}_t)) \tag{5}$$

Scaling factor 2 was chosen to expand the range of possible values taken by approximated probability. Empirical justification is presented in Figure 6. We evaluate scaling factors 0.5, 2, 10. The factor 2 yields the strongest separation between members and non-members.

The best performing method is our approach that utilizes the sigmoid function. It maps values to a bounded range, which reduces the impact of extreme perplexity outliers and prevents excessive compression during normalization. A comparison of the methods can be seen in Figure 7.

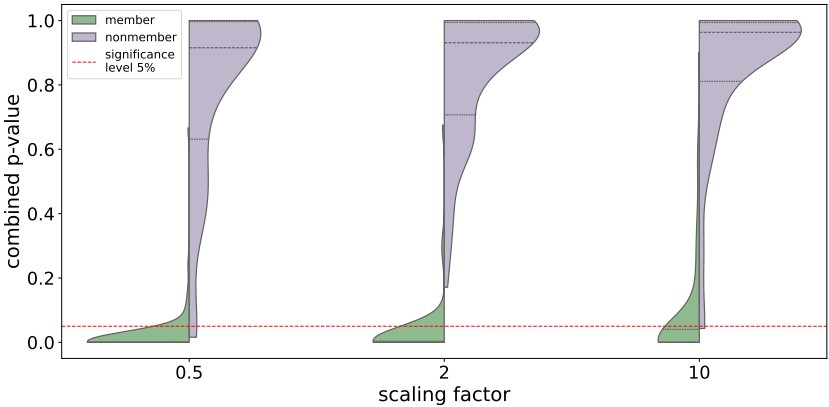

Figure 6: **Distribution of p-values.** We present the results across all Pile subsets on Pythia-12B-dedup for different scaling factors of sigmoid function.

A.7 NORMALIZATION AND TRIMMING OF EXTREME VALUES

Given a sequence of training metrics $\{t_j\}_{j=1}^n$ and corresponding validation metrics $\{v_j\}_{j=1}^m$, the Z-score normalization using only training statistics proceeds as follows:

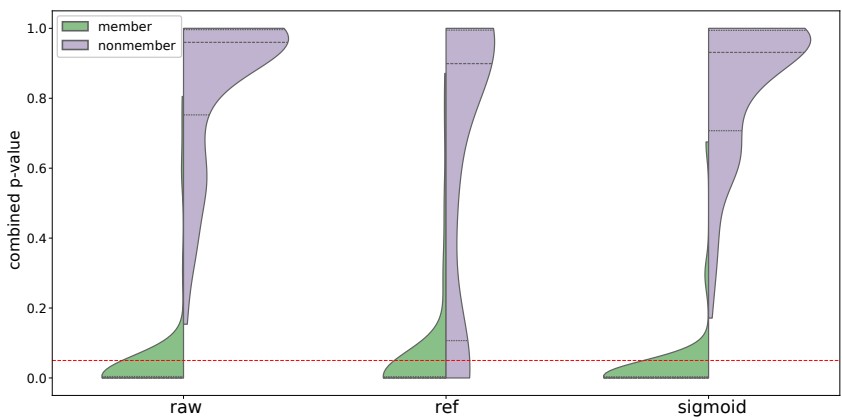

Figure 7: **Distribution of p-values Across all Pile Subsets on Pythia-12B-dedup.** For each subset, we run the scoring model 40 times with 1000 samples per run. The resulting p-values are grouped in batches of four and combined within each group. The violin plots display the distribution of these combined p-values aggregated from all subsets for three different methods: raw similarities, reference model (based on PETAL), and our approach based on the sigmoid function. The better method has lower p-values for members (shown in green) and higher p-values for non-members (shown in purple).

Compute the empirical mean and standard deviation of the training metrics:

$$\mu_t = \frac{1}{n}\sum_{j=1}^{n} t_j, \tag{6}$$

$$\sigma_t = \sqrt{\frac{1}{n}\sum_{j=1}^{n}\left(t_j - \mu_t\right)^2}. \tag{7}$$

Normalize both training and validation metrics using the training set statistics:

$$\tilde{t}_j = \frac{t_j - \mu_t}{\sigma_t}, \quad j = 1, \ldots, n, \tag{8}$$

$$\tilde{v}_k = \frac{v_k - \mu_t}{\sigma_t}, \quad k = 1, \ldots, m. \tag{9}$$

Here, $\tilde{t}_j$ and $\tilde{v}_k$ denote the normalized training and validation metric values, respectively.

The normalized training metrics satisfy $\mathbb{E}[\tilde{t}_j] = 0$, $\mathrm{Var}[\tilde{t}_j] = 1$.

Next, we trim extreme values by removing the 2.5% smallest and 2.5% largest values and replace them with the global mean.

### A.8 TRAINING OF THE LINEAR MODEL (FOR P-VALUES ONLY)

Next, we perform the following procedure for multiple random shuffles:

1. Shuffle the training and validation metric vectors independently.
2. Split the shuffled data into a training subset and a held-out subset of equal size.
3. Fit a linear model on the training subset to learn feature weights.
4. Apply the trained model to the held-out subset and conduct a one-sided t-test on the predicted scores to obtain a p-value.

Because the different held-out subsets overlap, the resulting p-values are statistically dependent. To aggregate these p-values $p_1, p_2, \ldots, p_n$ into a single combined p-value $p_{\text{combined}}$, we use the

Brown–Sidak correction:

$$p_{\text{combined}} = 1 - \exp\left(\sum_{i=1}^{n} \log(1 - p_i)\right).$$

This method is conservative and helps control the Type I error rate when tests are dependent.

### A.9 P-VALUE TESTS ACROSS MODEL SIZES

In order to assess differences across model sizes, we perform one-sided t-tests, from which we derive the corresponding p-values, as presented in Figure 8. These p-values are shown both to enable comparison with e-values and to provide consistency with prior work on dataset inference, where p-values were commonly employed.

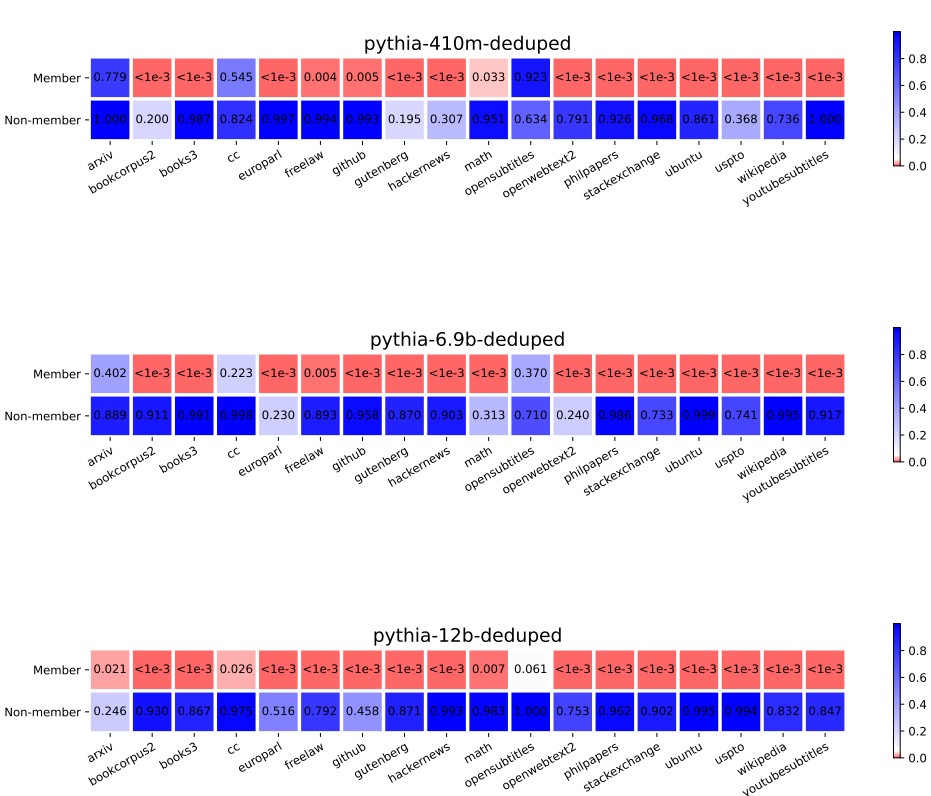

Figure 8: P-value distributions across different Pythia model sizes

## B IMPLEMENTATION DETAILS FOR BASELINE METHODS

We consider three baseline methods for the black-box LLM training data detection. Each baseline reflects a distinct notion of membership signal, quantified through a dedicated approach to measure the different in distribution between the suspect and held-out sets.

We use the following baseline methods:

1. **Baseline.** This is a naïve baseline based memorization of training data by LLMs. This method evaluates the degree to which an LLM reproduces the ground-truth continuations of given sequences. Specifically, model-generated suffixes are compared to the original suffixes using the RoBERTa

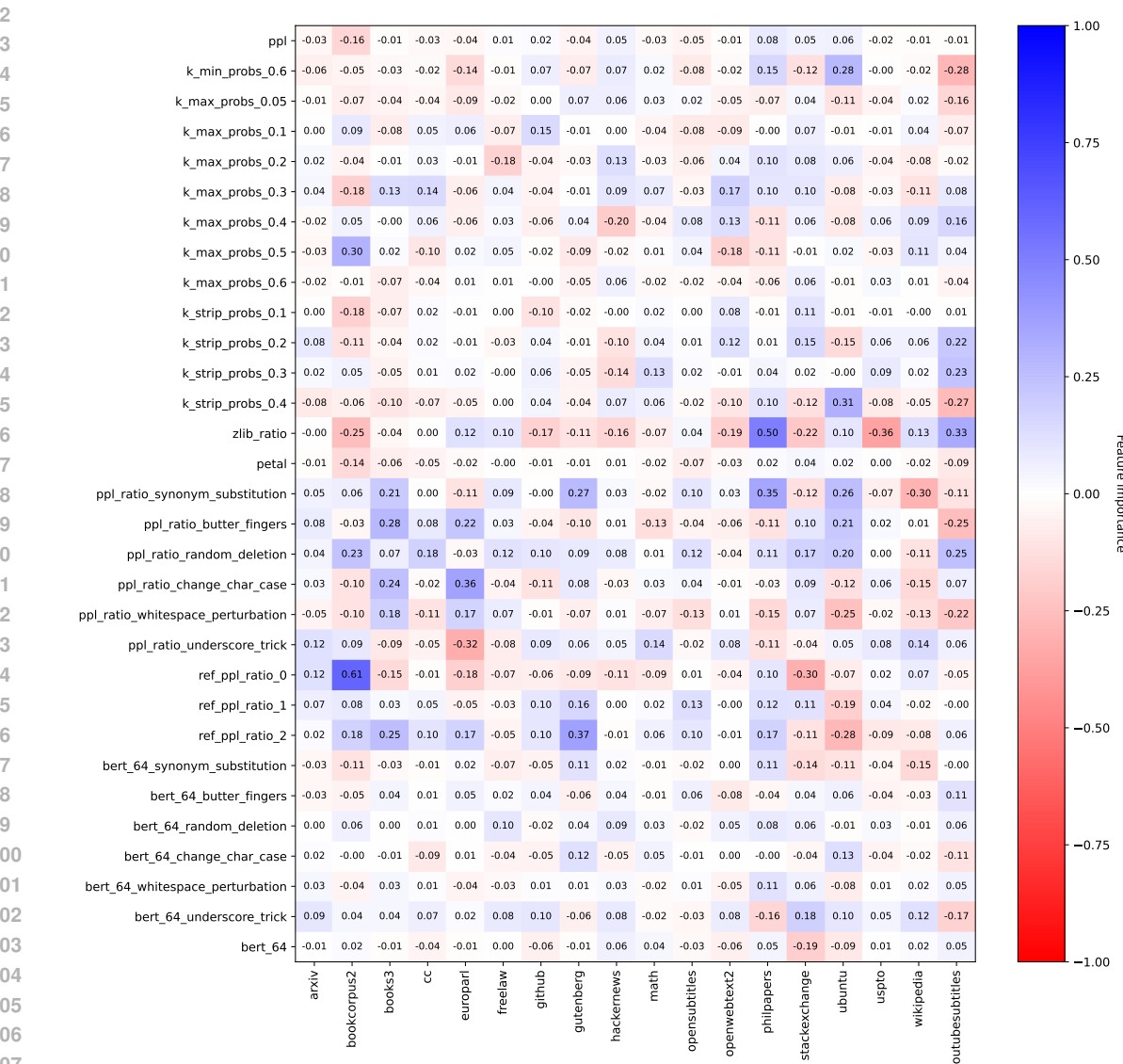

Figure 9: Member vs. non-member feature importance.

score, a variant of BERTScore (Zhang et al., 2020) in which the RoBERTa-large model (Liu et al., 2019) is employed. Higher similarity values are interpreted as stronger evidence of memorization.

2. **PETAL.** (He et al., 2025) For each example, PETAL estimates token-level probabilities by measuring the semantic similarity between model-generated and ground-truth tokens using reference model. These similarity scores are used to train linear regression model to map semantic similarities to log-probabilities, which are then aggregated at the sentence level. The resulting values are compared across suspicious and held-out sets.

3. **CatShift.** (Xiong et al., 2025) leverages the phenomenon of catastrophic forgetting, whereby models overwrite previously learned representations during fine-tuning but can also "reactivate" them when re-exposed to familiar data. What CatShift does, it splits a suspicious dataset into a training subset and a testing subset, then fine-tunes a target LLM on the training portion using a text completion task. Model outputs on the testing subset are collected both before and after fine-tuning, and their similarity is quantified using metrics such as BERT-based scores. The distributions of these similarity scores are then compared against those obtained from a known non-member validation set using statistical tests such as the Kolmogorov–Smirnov test. If the

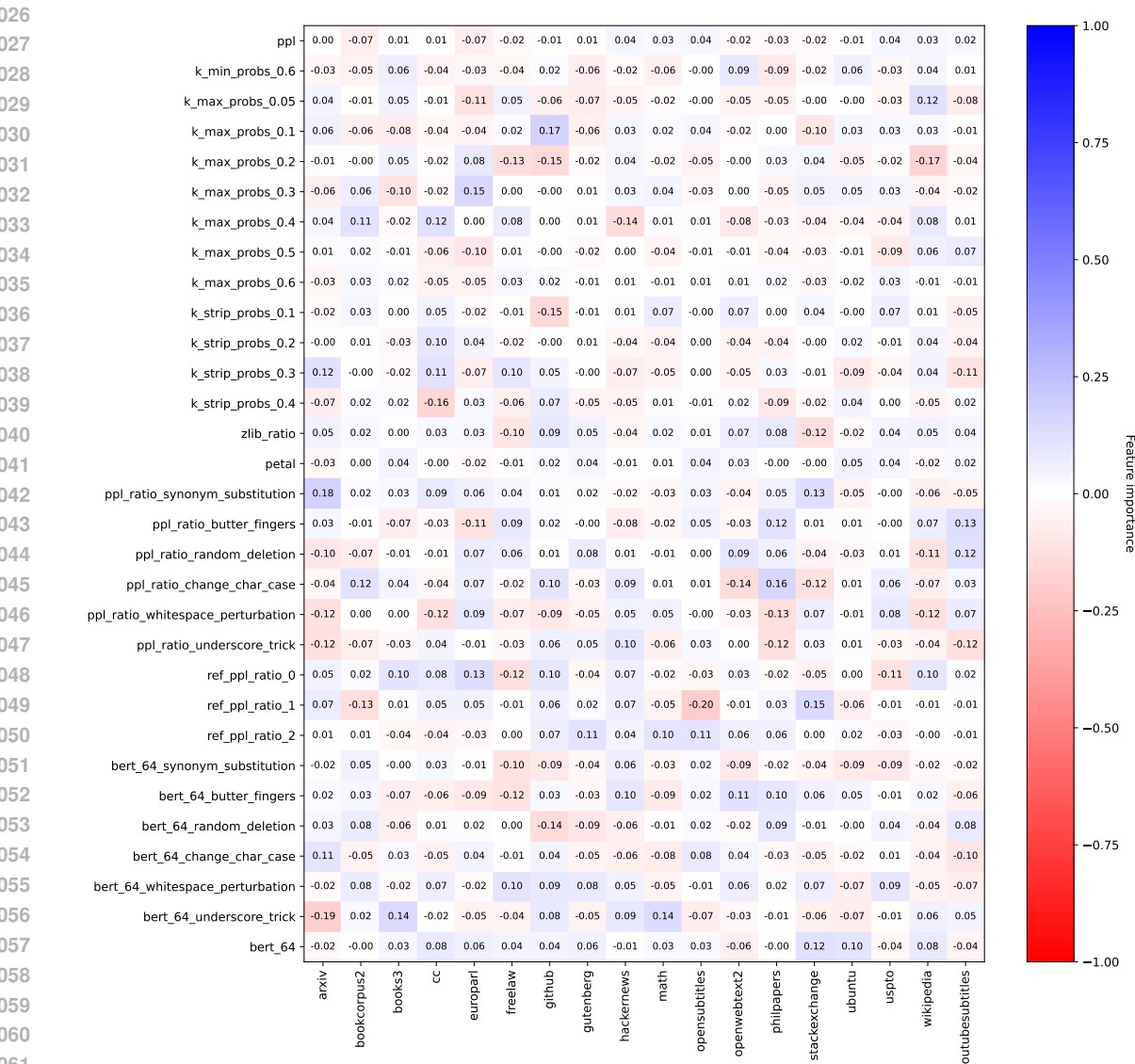

Figure 10: Non-member vs. non-member feature importance.

suspicious dataset was included in training, fine-tuning induces a stronger output shift, reflecting the reactivation of forgotten knowledge, whereas novel data induce only modest changes.

## B.1 BASELINE: NAÏVE APPROACH BASED ON MEMORIZATION

Each test example is split into a *prefix* (448 tokens) and a *suffix* (last 64 tokens). Then, we prompt the target model with *prefix* and generate exactly 64 tokens as the suffix. The generated prediction is compared to the original *suffix* by computing the RoBERTa score (using both the generated suffix and the ground-trugh one). Finally, we perform a one-sided t-test to assess whether the distribution of RoBERTa scores on the suspicious set differs significantly from that on the held-out set. P-values across data subsets used for this baseline method are presented in Figure 11.

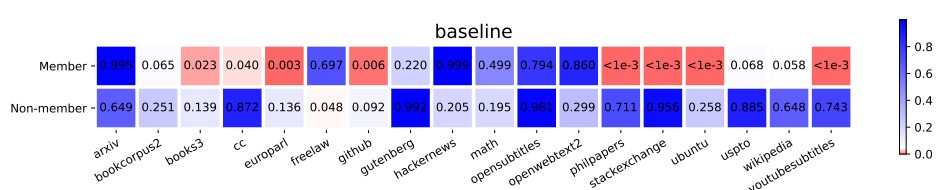

Figure 11: P-value distribution for the Baseline method on Pythia-12B-deduped.

## B.2 PETAL METHOD

The PETAL method (He et al., 2025) estimates next-token probabilities under black-box access using a surrogate model $M_s$. For each token position $t$, let the prefix be

$$p = x_1, x_2, \ldots, x_{t-1},$$

and let $x_t$ be the ground-truth next token. We query $M_s$ with:

```
--max-new-tokens=1
```

to obtain the surrogate prediction $\hat{x}_t = M_s(p)$ and log-probability $\log P_s(x_t \mid p)$. We then embed both $x_t$ and $\hat{x}_t$ using `sentence-transformers/all-MiniLM-L6-v2`, yielding vectors $\mathbf{e}(x_t)$ and $\mathbf{e}(\hat{x}_t)$. The cosine similarity is computed as

$$\text{sim}(x_t, \hat{x}_t) \;=\; \frac{\mathbf{e}(x_t) \cdot \mathbf{e}(\hat{x}_t)}{\|\mathbf{e}(x_t)\|\|\mathbf{e}(\hat{x}_t)\|}.$$

We fit a linear calibration function $f$ by regressing the surrogate log-probabilities onto these similarities:

$$\log P_s(x_t \mid p) \;=\; \alpha \log\big(\text{sim}(x_t, \hat{x}_t)\big) + \beta,$$

where $\alpha, \beta$ are learned via least squares.

To apply PETAL to the target model $M_t$, we prompt $M_t$ on the same prefix $p$ to generate $\hat{x}'_t = M_t(p)$, compute $\text{sim}(x_t, \hat{x}'_t)$, and estimate its log-probability as

$$\log \tilde{P}_t(x_t \mid p) \;=\; \alpha \log\big(\text{sim}(x_t, \hat{x}'_t)\big) + \beta.$$

For each example in both the suspect set $D_{\text{sus}}$ and the held-out set $D_{\text{hold}}$, we aggregate the estimated log-probabilities:

$$\bar{\ell} \;=\; \frac{1}{T} \sum_{t=1}^{T} \log \tilde{P}_t(x_t \mid x_{<t}).$$

Finally, a one-sided $t$-test is conducted on the distributions of $\bar{\ell}$ over $D_{\text{sus}}$ versus $D_{\text{hold}}$, yielding a $p$-value for each epoch. The resulting distribution of $p$-values is shown in Figure 12.

## B.3 CATSHIFT METHOD

We use the `sus` dataset comprising 1000 samples, evenly split into a fine-tuning set $D^{\text{train}}$ and an evaluation set $D^{\text{test}}$, each containing 500 samples. The base model $M_{\text{base}}$ is fine-tuned on $D^{\text{train}}$ for 10 epochs to yield $M_{\text{ft}}$. A held-out set $D^{\text{hold}}$ of 500 samples—drawn from the same data distribution but never used during the original training of $M_{\text{base}}$.

For each prefix–suffix pair $(p, s)$, where the suffix $s$ consists of the subsequent 64 tokens, both $M_{\text{base}}$ and $M_{\text{ft}}$ are prompted with:

```
--max-new-tokens=64
```

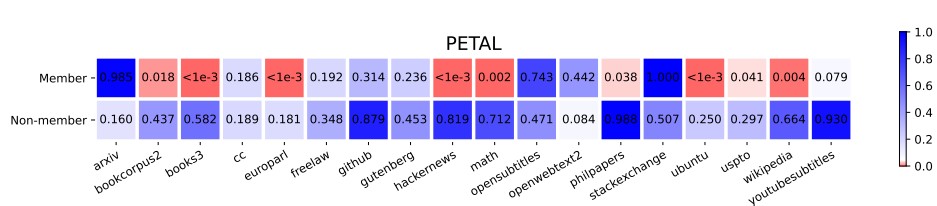

Figure 12: P-value distribution for PETAL method on Pythia-12B-deduped

and their responses are recorded. Let $y_{\text{base}} = M_{\text{base}}(p)$ and $y_{\text{ft}} = M_{\text{ft}}(p)$. We compute the BERT similarity $\text{BERT}(y_{\text{base}}, y_{\text{ft}})$ for each sample in $D^{\text{test}}$ and $D^{\text{hold}}$. To determine whether fine-tuning induces a statistically significant shift in generation behavior, we conduct a one-sided $t$-test on the distributions of BERT similarities from $D^{\text{test}}$ versus $D^{\text{hold}}$, producing a $p$-value at each epoch. The resulting $p$-values are shown in Figure 13 indicate a very poor performance (only one true positive after 1st or 3rd epoch and up to three after 10th epoch compared to 8 true positives for the simple Baseline).

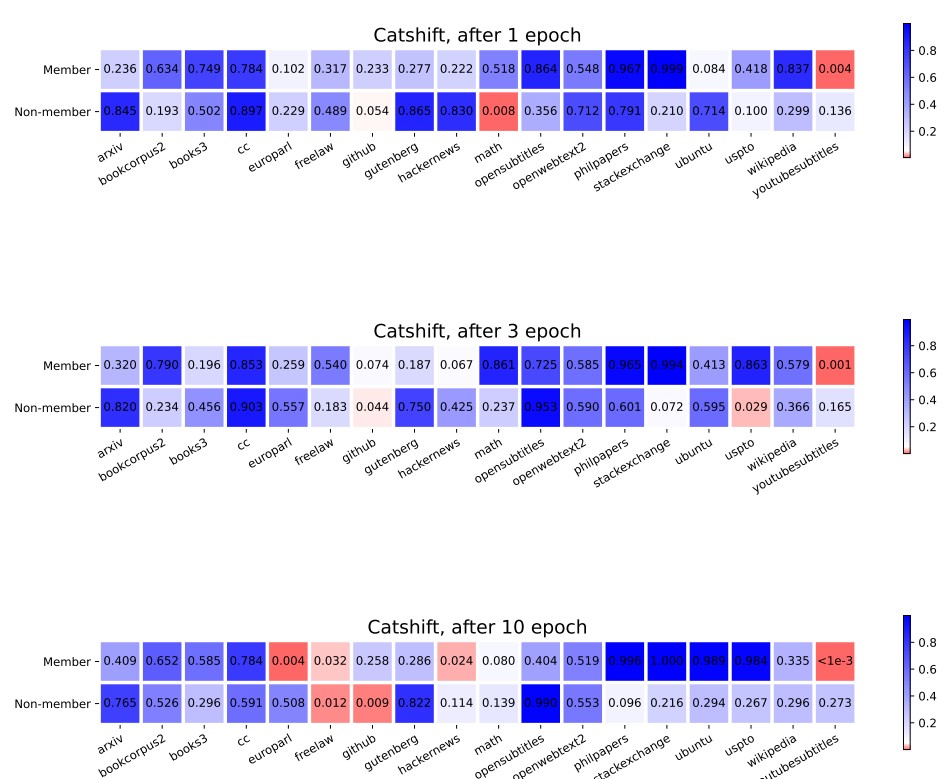

Figure 13: **Epoch-wise $p$-values for the CatShift Method.** We show the values after fine-tuning Pythia-410M-deduped (after 1st, 3rd, and 10th epoch). Similar trends are observed across intermediate epochs.

# C  NONPARAMETRIC TWO-SAMPLE TESTING BY BETTING

An *e-value* is a nonnegative statistic whose expectation under the null hypothesis is at most one. Intuitively, it quantifies evidence against the null: large e-values suggest the null is implausible, while small values are consistent with it. Following the betting framework of Shafer (2021); Shekhar & Ramdas (2023), hypothesis testing can be viewed as a *betting game* against the null. At each round $t$, the arbitrator observes a pair of data points $Z_t = (X_t, Y_t)$ and selects a payoff function $S_t$ with null-conditional expectation at most one. The bettor's wealth is updated multiplicatively as

$$W_t = W_{t-1} S_t(Z_t), \quad W_0 = 1. \tag{10}$$

The resulting sequence $\{W_t\}_{t \geq 0}$ is a nonnegative martingale under $H_0$, known as an *e-process* (Ramdas et al., 2020; Ramdas & Wang, 2025). By Ville's inequality (Ville, 1939), exceeding a threshold (e.g., $W_t \geq 1/\alpha$) guarantees a valid level-$\alpha$ test at any stopping time. Thus, the current wealth $W_t$ itself serves as an e-value, providing a calibrated measure of evidence against the null hypothesis. Such approach introduces numerous advantages over traditional p-value based testing, in particular intuitive properties allowing for simple interpretation of the results.

## C.1  BETTING STRATEGIES

Similarly to how there are better and worse estimators, the betting test may be better or worse based on the employed *betting strategy*. A betting strategy is a predictable, data-driven rule that generates stakes from past data, i.e., dictates the fraction of wealth that we bet in each round, based on our previous experience in the game.

We formalize past information via the filtration $(\mathcal{F}_t)_{t \geq 0}$, where for the sequence of data points $\{Z_t\}_{t \geq 1}$ (in our two-sample setting, $Z_t = (X_t, Y_t)$) we set filtration $\mathcal{F}_t = \sigma(Z_1, \ldots, Z_t)$ given by a $\sigma$-algebra on $\{Z_i\}_{i \leq t}$, and take trivial $\mathcal{F}_0$. A rule is *predictable* if it is $\mathcal{F}_{t-1}$-measurable.

Let $\mathcal{Z} = \mathcal{X} \times \mathcal{X}$ denote the observation space. A betting strategy is a sequence of $\mathcal{F}_{t-1}$-measurable maps

$$\mathcal{A}_{\text{bet}} = \{\lambda_t : \mathcal{Z}^{t-1} \to [-\lambda_{\max}, \lambda_{\max}]\}_{t \geq 1}.$$

Given a predictable score function $\tilde{g}_t$, we define the (scalar) round-$t$ score

$$u_t := \tilde{g}_t(Z_t) \in [-1, 1],$$

chosen so that under $H_0$, $\boldsymbol{E}[u_t \mid \mathcal{F}_{t-1}] \leq 0$.

In general, any payoff $S_t$ with $\boldsymbol{E}[S_t \mid \mathcal{F}_{t-1}] \leq 1$ yields valid e-values. A standard and convenient choice is the linearized form

$$S_t(Z_t) = 1 + \lambda_t u_t,$$

which cleanly separates *what* we bet on ($u_t$; produced by the prediction/witness rule) from *how much* we bet ($\lambda_t$; produced by the betting strategy). Linearized form ensures that $S_t \geq 0$ when $|u_t| \leq 1$ and $|\lambda_t| \leq 1$, and simplifies analysis and implementation.

**Log-Optimal Motivation and Stopping Rule.** A good betting strategy is a one that ensures quick growth of evidence under the alternative. This can be quantified by log-optimal principles (Kelly, 1956) and their modern developments (Shafer, 2021; Waudby-Smith & Ramdas, 2024; Grünwald et al., 2024). Motivated by those principles, we focus on strategies that approximately maximize expected log-wealth. Such strategies are conservative in the sense that they never stake all capital and therefore avoid ruin (zero wealth). The resulting wealth process $W_t^\star$ is a (test) martingale under $H_0$ and grows at an exponential rate under fixed alternatives. We adopt the stopping time

$$\tau := \inf\{t \geq 1 : W_t \geq 1/\alpha\},$$

which, by Ville's inequality (Ville, 1939), induces a level-$\alpha$ sequential test.

Exponential wealth growth under fixed alternatives is particularly valuable for dataset inference, where evidence may need to be established from few examples (e.g., in legal settings) or when computing many membership features is costly. When an effect is present, the expected log-wealth grows approximately linearly in $t$, so the wealth $\{W_t\}$ increases at an exponential rate. Consequently, the stopping time $\tau = \inf\{t \geq 1 : W_t \geq 1/\alpha\}$ can be small, meaning fewer observations are required

to reach a level-$\alpha$ decision—while anytime validity is preserved by the betting construction and Ville's inequality.

**RKHS and the Witness.** Let $K : \mathcal{X} \times \mathcal{X} \to \mathbb{R}$ be a positive definite kernel. The reproducing kernel Hilbert space (RKHS) $\mathcal{H}_K$ is the completion of finite linear combinations of $\{K(x, \cdot) : x \in \mathcal{X}\}$ with inner product satisfying the reproducing property

$$g(x) \;=\; \langle g, \, K(x, \cdot)\rangle_{\mathcal{H}_K}, \qquad \langle K(x, \cdot), K(y, \cdot)\rangle_{\mathcal{H}_K} = K(x, y).$$

We use the RKHS unit ball $\{g : \|g\|_{\mathcal{H}_K} \le 1\}$ as the witness class underlying kernel MMD. In our sequential procedure, the prediction strategy maintains a predictable witness $g_t \in \mathcal{H}_K$; given the round-$t$ pair $Z_t = (X_t, Y_t)$, the per-round edge is

$$v_t \;=\; g_t(X_t) - g_t(Y_t) \;=\; \big\langle g_t, \, K(X_t, \cdot) - K(Y_t, \cdot)\big\rangle_{\mathcal{H}_K}.$$

**Kernel MMD.** Kernel maximum mean discrepancy (MMD) underlies a widely used batch two-sample test (Gretton et al., 2012). Sequential nonparametric counterparts exist (Balsubramani & Ramdas, 2016; Manole & Ramdas, 2023), and a concrete sequential kernel-MMD construction with strong guarantees is developed by Shekhar & Ramdas (2023, Sec. 4), which we adopt here as part of our prediction strategy.

We view two-sample distance through the lens of an integral probability metric. For distributions $P_X$ and $P_Y$ on $\mathcal{X}$ with $X \sim P_X$ and $Y \sim P_Y$,

$$d_{\mathcal{G}}(P_X, P_Y) \;=\; \sup_{g \in \mathcal{G}} \big(\boldsymbol{E}[g(X)] - \boldsymbol{E}[g(Y)]\big).$$

Let $K : \mathcal{X} \times \mathcal{X} \to \mathbb{R}$ be a positive definite kernel with reproducing kernel Hilbert space (RKHS) $\mathcal{H}_K$. Specializing $\mathcal{G}$ to the RKHS unit ball yields the *kernel MMD*:

$$\mathrm{MMD}(P_X, P_Y) \;=\; \sup_{\|g\|_{\mathcal{H}_K} \le 1} \big(\boldsymbol{E}[g(X)] - \boldsymbol{E}[g(Y)]\big) \;=\; \|\mu_{P_X} - \mu_{P_Y}\|_{\mathcal{H}_K},$$

where $\mu_P := \boldsymbol{E}[K(X, \cdot)] \in \mathcal{H}_K$ is the kernel mean embedding.

We update the witness online via projected averaging (a special case of projected gradient ascent):

$$g_{t+1} \;\leftarrow\; \Pi_{\|g\|_{\mathcal{H}_K} \le 1}\Big(g_t \;+\; \tfrac{1}{t}\big(K(X_t, \cdot) - K(Y_t, \cdot)\big)\Big), \qquad t \ge 1,$$

where $\Pi$ denotes projection onto the RKHS unit ball. The scalar $v_t$ (or a bounded version $u_t \in [-1, 1]$) then feeds the linearized payoff $S_t(Z_t) = 1 + \lambda_t u_t$ used in the wealth update.

**Kolmogorov–Smirnov Discrepancy.** We consider a transformation $T : \mathcal{Z} \to \mathcal{Z}$ such that the null distribution remains invariant under this transformation, whereas distributions under the alternative are altered by it. Formally, let $\mathcal{P}_{\mathrm{null}}$ and $\mathcal{P}_{\mathrm{alt}}$ denote disjoint classes of probability distributions on the observation space $\mathcal{Z}$. We assume

$$P = P \circ T^{-1} \quad \forall\, P \in \mathcal{P}_{\mathrm{null}}, \qquad P \ne P \circ T^{-1} \quad \forall\, P \in \mathcal{P}_{\mathrm{alt}}.$$

This invariance property allows us to frame the hypothesis testing problem in terms of a *discrepancy measure* between $P$ and its transformed version $P \circ T^{-1}$.

To quantify this discrepancy, we employ an Integral Probability Metric (IPM). Given a function class $G = \{g : \mathcal{Z} \to [-\tfrac{1}{2}, \tfrac{1}{2}]\}$ is defined as

$$d_G(P, P \circ T^{-1}) := \sup_{g \in G} \Big| \mathbb{E}_P[g(Z)] - \mathbb{E}_P[g(TZ)] \Big|. \tag{11}$$

The Kolmogorov–Smirnov (KS) discrepancy arises as a special case of equation 11 when $G$ is chosen to be the class of indicator functions

$$G_{\mathrm{KS}} = \big\{\, g_u(x) = \mathbf{1}\{x \le u\} : u \in \mathbb{R} \,\big\}. \tag{12}$$

This yields the classical KS distance

$$d_{G_{\mathrm{KS}}}^{-}(P_X, P_Y) := \sup_{u \in \mathbb{R}} \big(F_X(u) - F_Y(u)\big). \tag{13}$$

where $F_X$ and $F_Y$ denote the cumulative distribution functions of $P_X$ and $P_Y$, respectively. In our hypothesis test of interest, we compare the distribution of the *suspect set* with that of a *held-out set*. We hypothesize that, under the alternative, the suspect set exhibits *smaller membership feature values* than the held-out set. This corresponds to a stochastic ordering in which the empirical CDF of the suspect set dominates that of the held-out set. Accordingly, we employ a one-sided Kolmogorov–Smirnov discrepancy and bet on maximizing the difference stated in equation 12.

**Online Newton Step (Staking).** At round $t$, the stake $\lambda_t$ is chosen predictably from past data (i.e., $\mathcal{F}_{t-1}$-measurable). After observing $Z_t$, we compute the bounded score $u_t \in [-1, 1]$ (from the kernel-MMD witness), update a one-dimensional curvature accumulator, and then set the next stake for round $t+1$:

$$
z_t = \frac{u_t}{1 + \lambda_t u_t}, \qquad a_t = a_{t-1} + z_t^2, \qquad \lambda_{t+1} = \Pi_{[-\lambda_{\max}, \lambda_{\max}]}\left(\lambda_t + \frac{2}{2 - \log 3}\frac{z_t}{a_t}\right),
$$

with initialization $\lambda_1 = 0$ and $a_0 = 1$. Here $\Pi_{[-\lambda_{\max}, \lambda_{\max}]}$ denotes projection onto the stake set.

**Kelly Betting**. Another approach to placing bets is the strategy introduced by (**?**), which aims to maximize expected log wealth by allocating bets in proportion to the strength of evidence for each outcome. This principled allocation leads to the highest achievable long-term exponential growth of wealth. The update rule given in equation 14 is quadratic approximation of the Kelly criterion.

$$
\lambda_t = \text{clip}\left(\frac{\sum_{j=1}^{t} u_j}{\sum_{j=1}^{t} u_j^2 + \varepsilon}, -\lambda_{\max}, \lambda_{\max}\right), \qquad t = 1, \ldots, N - 1. \tag{14}
$$

## C.2 ANYTIME VALIDITY AND POST-HOC SIGNIFICANCE.

Let $\mathcal{P}_0 = \{(P_X, P_Y) : P_X = P_Y\}$ denote the composite null. For any (possibly data-dependent) stopping time $\tau$ adapted to the data filtration and for any predictable strategies used to form the wealth process $W_t$, the e-process property and Ville's inequality imply

$$
\sup_{(P_X, P_Y) \in \mathcal{P}_0} \mathbb{P}_{P_X \times P_Y}\left(\sup_{t \geq 1} W_t \geq 1/\alpha\right) \leq \alpha.
$$

Equivalently, the test that rejects at $\tau = \inf\{t \geq 1 : W_t \geq 1/\alpha\}$ satisfies

$$
\sup_{(P_X, P_Y) \in \mathcal{P}_0} \mathbb{P}_{P_X \times P_Y}(\tau < \infty) \leq \alpha,
$$

so the probability of a false rejection is at most $\alpha$ *no matter when we choose to stop*. Formally, this uniform type-I control is established in Shekhar & Ramdas (2023, Thm. 1).

A key advantages of our framework are *anytime validity* and *post-hoc significance*. As discussed, the arbitrator may monitor the test sequentially and reject the null as soon as $W_t \geq 1/\alpha$ *without inflating type-I error*. By contrast, a classical fixed-horizon $p$-value is valid only for a pre-specified sample size (or stopping rule); repeatedly "peeking" at the data and continuing until $p \leq \alpha$ can inflate type-I error unless one uses explicit sequential corrections (e.g., $\alpha$-spending/group-sequential methods). Moreover, value of $W_t$ can be interpreted regardless of the predefined threshold, meaning that if the evidence extensively exceeded the threshold the arbitrator can claim more significant result or if the evidence did not reach initially assumed level an arbitrator can still decide to reject the null but at lower significance.

In our setting, we therefore use e-values: they provide anytime-valid, post-hoc interpretable evidence and, under fixed alternatives, exhibit approximately linear growth of expected log-wealth (hence exponential growth of $W_t$), yielding strong sample efficiency.

We use a *linearized payoff* built from the per-round kernel-MMD witness score. Concretely, with a predictable witness $g_t \in \mathcal{H}_K$ we set $u_t = g_t(X_t) - g_t(Y_t)$ and $S_t(Z_t) = 1 + \lambda_t u_t$. The anytime-valid type-I control and the associated consistency guarantees for this construction follow from the general theory and its kernel-MMD specialization Shekhar & Ramdas, 2023, Thm. 1; Prop. 3, Sec. 4. The betting strategy chooses the stake $\lambda_t$ via the online Newton step (ONS), predictably with respect to the data filtration $(\mathcal{F}_t)$ and using only past observations.

### C.3 SPECIALIZATION TO DATASET INFERENCE: OUR CONTRIBUTIONS

We instantiate the nonparametric two-sample betting framework from Appendix C for our DI. In all experiments we use the degree-2 polynomial kernel

$$K(x, y) \ = \ (\gamma \, x^\top y + c)^2, \qquad \gamma = \tfrac{1}{p}, \ c = 1,$$

with $p$ feature dimension. This choice preserves sensitivity to *mean differences* via the linear term (enabled by $c = 1$), the primary signal of interest in DI, while still allowing second-order interactions. We pair this kernel-MMD witness with an ONS staking rule to set the stakes $\lambda_t$ predictably from past data.

**Inherited Guarantees and Implications.** Because our construction matches the conditions in Appendix C, all theoretical guarantees carry over: the resulting test is *anytime-valid* (we may stop at the first time $t$ with $W_t \geq 1/\alpha$ without inflating type-I error), and *post-hoc significant* (we may interpret the results regardless of the threshold), and under fixed alternatives the expected log-wealth grows approximately linearly in $t$ (hence $W_t$ grows exponentially). Practically, this yields strong sample efficiency when data are scarce or computing membership features is costly. Moreover, under $H_0$ the wealth remains near its initial value (the payoff has null-conditional expectation at most one), leading to low false positive rates which is an important property in legal or high-stakes DI settings.

Overall, this specialization, degree-2 kernel for mean sensitivity plus ONS staking, adapts the general sequential MMD test to DI while retaining anytime-valid type-I control and fast evidence accumulation when member/non-member distributions differ.

### C.4 COMPARISON OF BADI AGAINST BASELINE METHODS

We report Area Under the Curve (AUC) metric for our method and two baseline methods (PETAL, Baseline) across three Pythia models: Pythia-410M (Table 1), Pythia-6.9B (Table 2), and Pythia-12B (Table 3). BADI consistently outperforms baseline methods for the majority of Pile subsets. Scaling up target model improves detection. For Pythia-12B AUC is significantly above random guess (50%) for every subset. As show in Figure 14, BADI demonstrates superior performance.

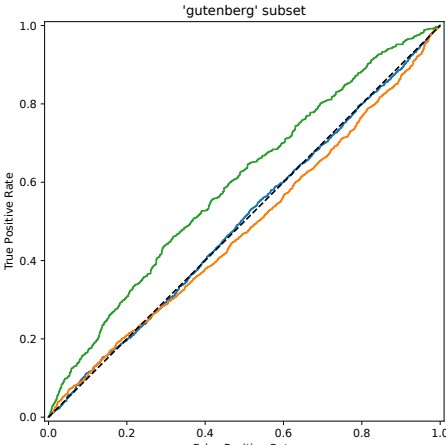
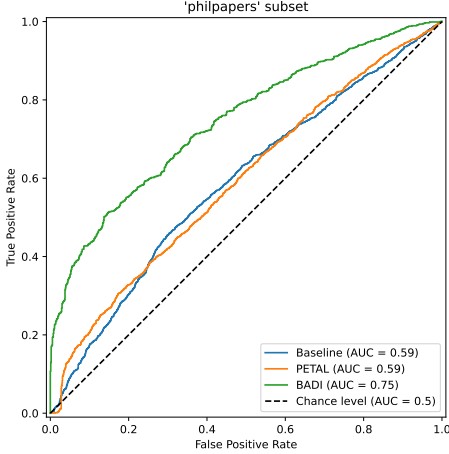

Figure 14: **ROC curves.** We present comparison of ROC curves for BADI, PETAL and Baseline for Pythia-12B model.

Table 1: AUC results of DI on Pile dataset and **Pythia-410M** model for BADI, PETAL and Baseline methods.

| Dataset | Baseline | PETAL | BADI (Ours) |
|---|---|---|---|
| ArXiv | **0.515** | 0.513 | 0.499 |
| BookCorpus2 | 0.503 | 0.512 | **0.593** |
| Books3 | 0.492 | 0.525 | **0.582** |
| Pile-CC | 0.513 | 0.514 | **0.519** |
| EuroParl | 0.457 | 0.441 | **0.635** |
| FreeLaw | 0.495 | 0.476 | **0.546** |
| Github | 0.529 | 0.511 | **0.558** |
| Gutenberg | 0.490 | 0.461 | **0.562** |
| HackerNews | 0.497 | 0.460 | **0.586** |
| DM Mathematics | 0.511 | **0.517** | 0.517 |
| OpenSubtitles | **0.503** | 0.493 | 0.492 |
| OpenWebText2 | 0.509 | 0.499 | **0.564** |
| Philpapers | 0.565 | 0.585 | **0.734** |
| StackExchange | 0.599 | 0.633 | **0.680** |
| Ubuntu IRC | 0.425 | 0.609 | **0.799** |
| USPTO | 0.511 | 0.529 | **0.602** |
| Wikipedia | 0.502 | 0.481 | **0.638** |
| YoutubeSubtitles | 0.423 | 0.365 | **0.728** |

Table 2: AUC results of DI on Pile dataset and **Pythia-6.9B** model for BADI, PETAL and Baseline methods.

| Dataset | Baseline | PETAL | BADI (Ours) |
|---|---|---|---|
| ArXiv | 0.510 | 0.520 | **0.536** |
| BookCorpus2 | 0.502 | 0.528 | **0.614** |
| Books3 | 0.506 | 0.542 | **0.594** |
| cc | 0.503 | **0.520** | 0.504 |
| EuroParl | 0.474 | 0.452 | **0.618** |
| FreeLaw | 0.509 | 0.485 | **0.546** |
| Github | 0.528 | 0.517 | **0.554** |
| Gutenberg | 0.495 | 0.477 | **0.613** |
| HackerNews | 0.480 | 0.465 | **0.581** |
| DM Mathematics | 0.506 | 0.521 | **0.565** |
| OpenSubtitles | 0.499 | 0.495 | **0.511** |
| OpenWebText2 | 0.494 | 0.497 | **0.551** |
| PhilPapers | 0.573 | 0.588 | **0.747** |
| StackExchange | 0.604 | 0.642 | **0.689** |
| Ubuntu IRC | 0.419 | 0.614 | **0.799** |
| USPTO | 0.507 | 0.532 | **0.591** |
| Wikipedia | 0.496 | 0.477 | **0.602** |
| YoutubeSubtiles | 0.424 | 0.369 | **0.700** |

# D  MORE RESULTS FOR TESTING BY BETTING

## D.1  ANYTIME VALIDITY

We increased the sequential horizon from 1000 data points to 2000 to test mean differences in membership features (members vs. non-members). This extension is valid by anytime-validity: the e-process allows continued monitoring without inflating type-I error, and we reject at the first $t$ with

Table 3: AUC results of DI on Pile dataset and **Pythia-12B** model for BADI, PETAL and Baseline methods.

| Dataset | Baseline | PETAL | BADI (Ours) |
|---|---|---|---|
| ArXiv | 0.512 | 0.522 | **0.541** |
| BookCorpus2 | 0.510 | 0.529 | **0.605** |
| Books3 | 0.508 | 0.546 | **0.605** |
| Pile-CC | 0.513 | 0.522 | **0.539** |
| EuroParl | 0.476 | 0.456 | **0.623** |
| FreeLaw | 0.500 | 0.487 | **0.540** |
| Github | 0.526 | 0.518 | **0.548** |
| Gutenberg | 0.500 | 0.481 | **0.594** |
| HackerNews | 0.477 | 0.466 | **0.573** |
| DM Mathematics | 0.503 | **0.523** | 0.520 |
| OpenSubtitles | 0.509 | 0.500 | **0.526** |
| Open Web Text2 | 0.506 | 0.500 | **0.536** |
| PhilPapers | 0.588 | 0.592 | **0.745** |
| StackExchange | 0.598 | 0.644 | **0.687** |
| Ubuntu IRC | 0.419 | 0.618 | **0.821** |
| USPTO | 0.519 | 0.533 | **0.573** |
| Wikipedia | 0.494 | 0.477 | **0.616** |
| YoutubeSubtitles | 0.427 | 0.370 | **0.717** |

$W_t \geq 1/\alpha$ (here $\alpha = 0.05$). Several PILE subsets with low TPR at 1000 data points (Table 4) improve markedly when more data points are available, for example, `hackernews` increases from 0.20 to 0.73. The longer horizon also uncensors runs that did not cross by 1000, allowing us to report additional stopping times; consequently, the mean reported stop time increases, reflecting the inclusion of late (but valid) rejections.

As shown in Figure 15, we plot mean ($\pm 95\%$ CI (Confidence Intervals)) wealth trajectories for the `hackernews` subset. With 1000 data points, the mean wealth remains below the $1/\alpha$ threshold, so only a minority of trials reject the null hypothesis. Increasing to 2000 data points yields substantially more threshold crossings; among rejecting runs, the average stopping time is about 948 observations. By anytime validity, extending the horizon in this way does not inflate type-I error.

A note on false positives in Figure 15. To estimate the false positive rate, we compare non-member vs. non-member streams. Due to limited non-member data, these runs were truncated at 1000 points. This truncation does not undermine our FPR conclusions: under the null, the wealth process is a nonnegative test martingale with $\mathbb{P}(\sup_t W_t \geq 1/\alpha) \leq \alpha$, so wealth does not systematically exceed its initial value and type-I error remains controlled. Empirically, we observe wealth staying near one and rare threshold crossings across datasets, consistent with this guarantee.

### D.2 ADDITIONAL INFORMATION ON GENERALIZATION TO DIFFERENT MODEL SIZES

We report performance metrics for our dataset inference framework across three Pythia models: Pythia-410M (Table 5), Pythia-6.9B (Table 7), and Pythia-12B (Table 6). The FPR matches the theoretical guarantee, remaining at 1% across datasets and model sizes. Scaling up improves detection on Books3, BookCorpus2, and Gutenberg, reflected by higher TPR and shorter average stopping times. Detection is consistently strong on Ubuntu IRC, StackExchange, PhilPapers, and YouTube Subtitles, with high TPR and early stopping. In contrast, ArXiv, Pile-CC, OpenSubtitles, FreeLaw, and GitHub rarely reach significance within the available 2000 examples. Scale effects are not uniform: for Wikipedia (and in some cases EuroParl and USPTO), larger models yield a weaker signal (lower TPR) despite similar FPR.

### D.3 ONE-SIDED TESTING

In dataset inference, one natural hypothesis is that, under the alternative, the suspect set exhibits *smaller membership feature values* than the held-out reference set. To evaluate this hypothesis, we employ the Kolmogorov–Smirnov (KS) discrepancy as our distance metric and place one-sided bets

Table 4: Performance metrics for sequential tests across datasets using membership features of our proposed method and **Pythia-410M** model. We have performed the test across datasets from the Pile corpus with **1000 member/non-member** data points with 5% significance level. Here N/A indicates that the test has failed to reach the significance level of 5% within range of 1000 data points.

| Dataset | TPR | FPR | FNR | Avg. stop time |
|---|---|---|---|---|
| ArXiv | $0.01 \pm 0.00$ | $0.01 \pm 0.01$ | $0.99 \pm 0.00$ | N/A |
| BookCorpus2 | $0.30 \pm 0.28$ | $0.01 \pm 0.01$ | $0.70 \pm 0.28$ | N/A |
| Books3 | $0.26 \pm 0.24$ | $0.01 \pm 0.02$ | $0.74 \pm 0.24$ | N/A |
| Pile-CC | $0.01 \pm 0.01$ | $0.01 \pm 0.00$ | $0.99 \pm 0.01$ | N/A |
| EuroParl | $0.60 \pm 0.20$ | $0.01 \pm 0.01$ | $0.40 \pm 0.20$ | $652.32 \pm 110.52$ |
| FreeLaw | $0.01 \pm 0.01$ | $0.01 \pm 0.00$ | $0.99 \pm 0.01$ | N/A |
| Github | $0.02 \pm 0.02$ | $0.01 \pm 0.02$ | $0.98 \pm 0.02$ | N/A |
| Gutenberg | $0.02 \pm 0.02$ | $0.01 \pm 0.01$ | $0.98 \pm 0.02$ | N/A |
| HackerNews | $0.20 \pm 0.13$ | $0.01 \pm 0.00$ | $0.80 \pm 0.13$ | N/A |
| DM Mathematics | $0.01 \pm 0.01$ | $0.01 \pm 0.02$ | $0.99 \pm 0.01$ | N/A |
| OpenSubtitles | $0.01 \pm 0.00$ | $0.02 \pm 0.04$ | $0.99 \pm 0.00$ | N/A |
| OpenWebText2 | $0.03 \pm 0.04$ | $0.01 \pm 0.00$ | $0.97 \pm 0.04$ | N/A |
| PhilPapers | $0.94 \pm 0.07$ | $0.01 \pm 0.01$ | $0.06 \pm 0.07$ | $175.57 \pm 67.93$ |
| StackExchange | $0.94 \pm 0.06$ | $0.01 \pm 0.01$ | $0.06 \pm 0.06$ | $305.02 \pm 69.95$ |
| Ubuntu IRC | $1.00 \pm 0.00$ | $0.02 \pm 0.02$ | $0.00 \pm 0.00$ | $92.32 \pm 21.32$ |
| USPTO | $0.48 \pm 0.24$ | $0.01 \pm 0.01$ | $0.52 \pm 0.24$ | $760.52 \pm 105.67$ |
| Wikipedia | $0.81 \pm 0.17$ | $0.01 \pm 0.01$ | $0.19 \pm 0.17$ | $461.40 \pm 126.33$ |
| YoutubeSubtitles | $0.98 \pm 0.04$ | $0.01 \pm 0.00$ | $0.02 \pm 0.04$ | $212.15 \pm 66.11$ |

Table 5: **Dataset Inference on the Pythia-410M Model.** Performance metrics for sequential tests across datasets using membership features of our proposed method and **Pythia-410M** model. We have performed the test across datasets from the Pile corpus with **2000 member/non-member** data points with 5% significance level. Here, N/A indicates that the test has failed to reach the significance level of 5% within range of 2000 data points.

| Dataset | TPR | FPR | FNR | Avg. stop time |
|---|---|---|---|---|
| ArXiv | $0.01 \pm 0.01$ | $0.01 \pm 0.01$ | $0.99 \pm 0.01$ | N/A |
| BookCorpus2 | $0.82 \pm 0.13$ | $0.01 \pm 0.01$ | $0.18 \pm 0.13$ | $638.62 \pm 188.59$ |
| Books3 | $0.78 \pm 0.19$ | $0.01 \pm 0.01$ | $0.22 \pm 0.19$ | $1070.20 \pm 263.09$ |
| Pile-CC | $0.01 \pm 0.00$ | $0.01 \pm 0.01$ | $0.99 \pm 0.00$ | N/A |
| EuroParl | $0.94 \pm 0.06$ | $0.01 \pm 0.00$ | $0.06 \pm 0.06$ | $477.05 \pm 144.63$ |
| FreeLaw | $0.02 \pm 0.01$ | $0.01 \pm 0.01$ | $0.98 \pm 0.01$ | N/A |
| Github | $0.10 \pm 0.07$ | $0.01 \pm 0.00$ | $0.90 \pm 0.07$ | N/A |
| Gutenberg | $0.34 \pm 0.19$ | $0.02 \pm 0.02$ | $0.66 \pm 0.19$ | $1645.88 \pm 150.15$ |
| HackerNews | $0.73 \pm 0.18$ | $0.01 \pm 0.01$ | $0.27 \pm 0.18$ | $948.67 \pm 246.53$ |
| DM Mathematics | $0.02 \pm 0.01$ | $0.01 \pm 0.00$ | $0.98 \pm 0.01$ | N/A |
| OpenSubtitles | $0.01 \pm 0.00$ | $0.01 \pm 0.01$ | $0.99 \pm 0.00$ | N/A |
| OpenWebText2 | $0.13 \pm 0.09$ | $0.01 \pm 0.01$ | $0.87 \pm 0.09$ | N/A |
| PhilPapers | $0.97 \pm 0.04$ | $0.01 \pm 0.02$ | $0.03 \pm 0.04$ | $194.60 \pm 96.15$ |
| StackExchange | $0.99 \pm 0.01$ | $0.01 \pm 0.00$ | $0.01 \pm 0.01$ | $192.19 \pm 51.38$ |
| Ubuntu IRC | $1.00 \pm 0.00$ | $0.01 \pm 0.01$ | $0.00 \pm 0.00$ | $81.37 \pm 12.23$ |
| USPTO | $0.85 \pm 0.14$ | $0.01 \pm 0.00$ | $0.15 \pm 0.14$ | $818.72 \pm 228.69$ |
| Wikipedia | $0.97 \pm 0.03$ | $0.02 \pm 0.02$ | $0.03 \pm 0.03$ | $320.27 \pm 107.87$ |
| YoutubeSubtitles | $1.00 \pm 0.00$ | $0.01 \pm 0.01$ | $0.00 \pm 0.00$ | $190.21 \pm 32.07$ |

on the event that the empirical CDF of the suspect set dominates that of the held-out set, as described in subsection C.1. The results of the one-sided hypothesis testing is consistent with the other test results presented in the study. We also observe an increase in the TPR of ArXiv, BookCorpus2 and Books3 subsets of The Pile dataset.

The test is implemented using the KS discrepancy together with the ONS betting strategy. Consistent with the other experiments in this study, we run the test with 50 random seeds and report the average

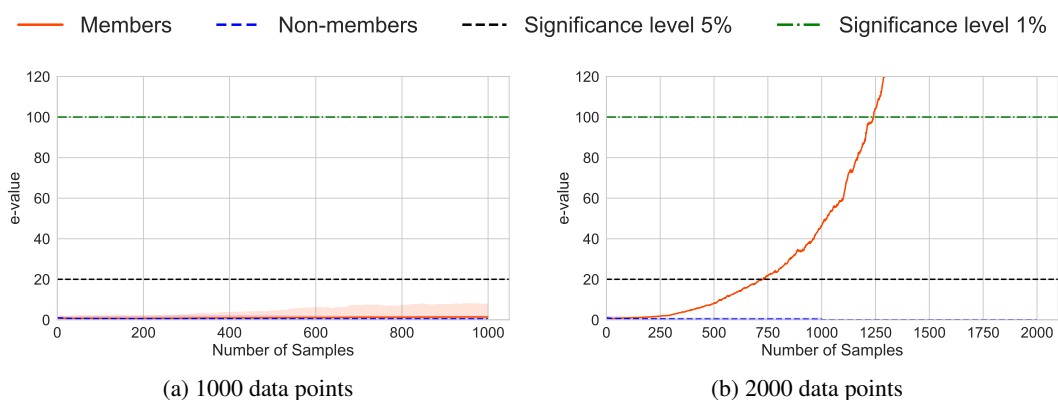

(a) 1000 data points          (b) 2000 data points

Figure 15: **Wealth Trajectories.** We present the wealth trajectories with 95% confidence intervals for `hackernews` (left) with test performed on 1000 data points and (right) test performed on 2000 data points of our dataset inference method and Pythia-410M model. Due to anytime validity of sequential testing, we can add extra data points to the test and eventually cross the desired threshold in the case of true positives.

Table 6: **Dataset Inference on the Pythia-12B Model.** Performance metrics of our DI approach across subsets of the Pile. Here, N/A indicates that the test has failed to reach the significance level of 5% within range of **2000 data points**.

| Dataset | TPR | FPR | FNR | Avg. stop time |
|---|---|---|---|---|
| ArXiv | $0.02 \pm 0.01$ | $0.01 \pm 0.01$ | $0.98 \pm 0.01$ | N/A |
| BookCorpus2 | $0.92 \pm 0.09$ | $0.01 \pm 0.01$ | $0.08 \pm 0.09$ | $532.64 \pm 178.19$ |
| Books3 | $0.85 \pm 0.15$ | $0.01 \pm 0.01$ | $0.15 \pm 0.15$ | $788.48 \pm 233.96$ |
| Pile-CC | $0.02 \pm 0.01$ | $0.01 \pm 0.01$ | $0.98 \pm 0.01$ | N/A |
| EuroParl | $0.68 \pm 0.19$ | $0.01 \pm 0.01$ | $0.32 \pm 0.19$ | $1021.10 \pm 249.76$ |
| FreeLaw | $0.01 \pm 0.01$ | $0.01 \pm 0.00$ | $0.99 \pm 0.01$ | N/A |
| Github | $0.09 \pm 0.06$ | $0.01 \pm 0.01$ | $0.91 \pm 0.06$ | N/A |
| Gutenberg | $0.60 \pm 0.19$ | $0.01 \pm 0.01$ | $0.40 \pm 0.19$ | $1222.45 \pm 224.66$ |
| HackerNews | $0.63 \pm 0.23$ | $0.01 \pm 0.01$ | $0.37 \pm 0.23$ | $1156.44 \pm 296.82$ |
| DM Mathematics | $0.03 \pm 0.02$ | $0.01 \pm 0.01$ | $0.97 \pm 0.02$ | N/A |
| OpenSubtitles | $0.01 \pm 0.01$ | $0.01 \pm 0.01$ | $0.99 \pm 0.01$ | N/A |
| OpenWebText2 | $0.01 \pm 0.00$ | $0.01 \pm 0.01$ | $0.99 \pm 0.00$ | N/A |
| PhilPapers | $0.97 \pm 0.07$ | $0.01 \pm 0.01$ | $0.03 \pm 0.07$ | $256.27 \pm 136.63$ |
| StackExchange | $0.98 \pm 0.03$ | $0.01 \pm 0.01$ | $0.02 \pm 0.03$ | $265.46 \pm 89.34$ |
| Ubuntu IRC | $1.00 \pm 0.00$ | $0.01 \pm 0.02$ | $0.00 \pm 0.00$ | $92.33 \pm 20.82$ |
| USPTO | $0.45 \pm 0.20$ | $0.01 \pm 0.01$ | $0.55 \pm 0.20$ | $1507.76 \pm 195.08$ |
| Wikipedia | $0.88 \pm 0.10$ | $0.01 \pm 0.01$ | $0.12 \pm 0.10$ | $649.32 \pm 180.11$ |
| YoutubeSubtitles | $0.96 \pm 0.08$ | $0.01 \pm 0.01$ | $0.04 \pm 0.08$ | $311.05 \pm 149.62$ |

TPR, FPR, and FNR at a significance level of $\alpha = 0.05$. The results for the Pythia-12B model are summarized in table 8.

### D.4 ABLATION ON BETTING STRATEGIES

We further evaluate the impact of different betting strategies on the performance of our sequential testing framework. In particular, we consider an approximate Kelly betting strategy, as described in subsection C.1. This strategy preserves the same theoretical guarantees as the ONS betting approach, and we empirically observe an exponential growth in wealth under this setup as well.

The experimental configuration mirrors the other experiments presented in this paper. To ensure consistency, we use Kernel-MMD as the distance measure and vary only the betting strategy. The results of sequential testing with the Kelly strategy for the Pythia-12B model are summarized in

Table 7: **Dataset Inference on the Pythia-6.9B Model.** Performance metrics of our DI approach across subsets of The Pile. Here, N/A indicates that the test has failed to reach the significance level of 5% within range of 2000 data points.

| Dataset | TPR | FPR | FNR | Avg. stop time |
|---|---|---|---|---|
| ArXiv | $0.01 \pm 0.01$ | $0.01 \pm 0.01$ | $0.99 \pm 0.01$ | N/A |
| BookCorpus2 | $0.95 \pm 0.04$ | $0.01 \pm 0.01$ | $0.05 \pm 0.04$ | $469.63 \pm 125.88$ |
| Books3 | $0.74 \pm 0.19$ | $0.01 \pm 0.01$ | $0.26 \pm 0.19$ | $979.44 \pm 283.20$ |
| Pile-CC | $0.01 \pm 0.01$ | $0.01 \pm 0.01$ | $0.99 \pm 0.01$ | N/A |
| EuroParl | $0.63 \pm 0.21$ | $0.01 \pm 0.01$ | $0.37 \pm 0.21$ | $1084.40 \pm 272.76$ |
| FreeLaw | $0.03 \pm 0.02$ | $0.01 \pm 0.01$ | $0.97 \pm 0.02$ | N/A |
| Github | $0.07 \pm 0.05$ | $0.01 \pm 0.01$ | $0.93 \pm 0.05$ | N/A |
| Gutenberg | $0.53 \pm 0.23$ | $0.01 \pm 0.01$ | $0.47 \pm 0.23$ | $1331.22 \pm 248.45$ |
| HackerNews | $0.56 \pm 0.21$ | $0.01 \pm 0.01$ | $0.44 \pm 0.21$ | $1298.19 \pm 249.46$ |
| DM Mathematics | $0.05 \pm 0.05$ | $0.01 \pm 0.01$ | $0.95 \pm 0.05$ | N/A |
| OpenSubtitles | $0.01 \pm 0.01$ | $0.01 \pm 0.01$ | $0.99 \pm 0.01$ | N/A |
| OpenWebText2 | $0.04 \pm 0.03$ | $0.01 \pm 0.01$ | $0.96 \pm 0.03$ | N/A |
| PhilPapers | $0.98 \pm 0.03$ | $0.01 \pm 0.01$ | $0.02 \pm 0.03$ | $221.95 \pm 75.00$ |
| StackExchange | $0.98 \pm 0.02$ | $0.02 \pm 0.06$ | $0.02 \pm 0.02$ | $258.49 \pm 80.11$ |
| Ubuntu IRC | $1.00 \pm 0.00$ | $0.01 \pm 0.01$ | $0.00 \pm 0.00$ | $105.64 \pm 33.86$ |
| USPTO | $0.47 \pm 0.21$ | $0.01 \pm 0.01$ | $0.53 \pm 0.21$ | $1465.17 \pm 204.61$ |
| Wikipedua | $0.93 \pm 0.06$ | $0.01 \pm 0.01$ | $0.07 \pm 0.06$ | $532.54 \pm 157.63$ |
| YoutubeSubtitles | $0.91 \pm 0.12$ | $0.01 \pm 0.01$ | $0.09 \pm 0.12$ | $379.51 \pm 202.74$ |

Table 8: **One-sided Hypothesis Testing for DI on the Pythia-12B Model.** Performance of our dataset inference (DI) method under a one-sided Kolmogorov–Smirnov testing setup across subsets of *the Pile* dataset. Results are reported at a significance level of $\alpha = 0.05$. N/A indicates that the test did not reach the significance threshold within 2000 data points.

| Dataset | TPR | FPR | FNR | Avg. stop time |
|---|---|---|---|---|
| ArXiv | $0.14 \pm 0.35$ | $0.00 \pm 0.00$ | $0.86 \pm 0.35$ | N/A |
| BookCorpus2 | $1.00 \pm 0.00$ | $0.02 \pm 0.14$ | $0.00 \pm 0.00$ | $166.98 \pm 112.58$ |
| Books3 | $1.00 \pm 0.00$ | $0.04 \pm 0.20$ | $0.00 \pm 0.00$ | $343.60 \pm 269.42$ |
| Pile-CC | $0.02 \pm 0.14$ | $0.02 \pm 0.14$ | $0.98 \pm 0.14$ | N/A |
| EuroParl | $1.00 \pm 0.00$ | $0.00 \pm 0.00$ | $0.00 \pm 0.00$ | $234.26 \pm 120.77$ |
| FreeLaw | $0.24 \pm 0.43$ | $0.04 \pm 0.20$ | $0.76 \pm 0.43$ | N/A |
| Github | $0.84 \pm 0.37$ | $0.04 \pm 0.20$ | $0.16 \pm 0.37$ | $1328.00 \pm 584.35$ |
| Gutenberg | $1.00 \pm 0.00$ | $0.02 \pm 0.14$ | $0.00 \pm 0.00$ | $272.64 \pm 159.49$ |
| HackerNews | $1.00 \pm 0.00$ | $0.04 \pm 0.20$ | $0.00 \pm 0.00$ | $543.68 \pm 315.14$ |
| DM Mathematics | $0.04 \pm 0.20$ | $0.02 \pm 0.14$ | $0.96 \pm 0.20$ | N/A |
| OpenSubtitles | $0.02 \pm 0.14$ | $0.00 \pm 0.00$ | $0.98 \pm 0.14$ | N/A |
| OpenWebText2 | $0.18 \pm 0.39$ | $0.02 \pm 0.14$ | $0.82 \pm 0.39$ | N/A |
| PhilPapers | $1.00 \pm 0.00$ | $0.02 \pm 0.14$ | $0.00 \pm 0.00$ | $43.76 \pm 28.15$ |
| StackExchange | $1.00 \pm 0.00$ | $0.04 \pm 0.20$ | $0.00 \pm 0.00$ | $77.74 \pm 54.74$ |
| Ubuntu IRC | $1.00 \pm 0.00$ | $0.00 \pm 0.00$ | $0.00 \pm 0.00$ | $25.14 \pm 14.77$ |
| USPTO | $1.00 \pm 0.00$ | $0.04 \pm 0.20$ | $0.00 \pm 0.00$ | $475.02 \pm 315.42$ |
| Wikipedia | $1.00 \pm 0.00$ | $0.04 \pm 0.20$ | $0.00 \pm 0.00$ | $242.14 \pm 135.59$ |
| YoutubeSubtitles | $1.00 \pm 0.00$ | $0.02 \pm 0.14$ | $0.00 \pm 0.00$ | $60.76 \pm 39.95$ |

table 9. Our findings indicate that, as long as the theoretical guarantees of the betting framework are preserved, we consistently observe exponential wealth growth, matching the results obtained with the ONS strategy reported in table 6.

## D.5 ABLATION ON TEST POWER

Table 9: **Ablation on Betting Strategies: Kelly Betting for DI on Pythia-12B.** Performance of our DI method using the approximate Kelly betting strategy across subsets of *The Pile*, with Kernel-MMD as the distance measure. Results are reported at a significance level of $\alpha = 0.05$. N/A indicates that the test did not reach the significance threshold within 2000 data points.

| Dataset | TPR | FPR | FNR | Avg. stop time |
|---|---|---|---|---|
| ArXiv | $0.03 \pm 0.02$ | $0.02 \pm 0.01$ | $0.97 \pm 0.02$ | N/A |
| BookCorpus2 | $0.95 \pm 0.07$ | $0.02 \pm 0.02$ | $0.05 \pm 0.07$ | $528.80 \pm 180.23$ |
| Books3 | $0.81 \pm 0.16$ | $0.02 \pm 0.01$ | $0.19 \pm 0.16$ | $852.12 \pm 224.82$ |
| Pile-CC | $0.03 \pm 0.02$ | $0.02 \pm 0.01$ | $0.97 \pm 0.02$ | N/A |
| EuroParl | $0.69 \pm 0.18$ | $0.02 \pm 0.02$ | $0.31 \pm 0.18$ | $1001.73 \pm 240.00$ |
| FreeLaw | $0.04 \pm 0.03$ | $0.02 \pm 0.01$ | $0.96 \pm 0.03$ | N/A |
| GitHub | $0.09 \pm 0.06$ | $0.02 \pm 0.02$ | $0.91 \pm 0.06$ | N/A |
| Gutenberg | $0.59 \pm 0.19$ | $0.03 \pm 0.03$ | $0.41 \pm 0.19$ | $1257.49 \pm 229.97$ |
| HackerNews | $0.55 \pm 0.23$ | $0.02 \pm 0.01$ | $0.45 \pm 0.23$ | $1293.87 \pm 265.74$ |
| DM Mathematics | $0.04 \pm 0.05$ | $0.02 \pm 0.01$ | $0.96 \pm 0.05$ | N/A |
| OpenSubtitles | $0.02 \pm 0.02$ | $0.02 \pm 0.02$ | $0.98 \pm 0.02$ | N/A |
| OpenWebText2 | $0.04 \pm 0.02$ | $0.02 \pm 0.02$ | $0.96 \pm 0.02$ | N/A |
| PhilPapers | $0.96 \pm 0.07$ | $0.02 \pm 0.02$ | $0.04 \pm 0.07$ | $243.87 \pm 139.10$ |
| StackExchange | $0.99 \pm 0.02$ | $0.02 \pm 0.01$ | $0.01 \pm 0.02$ | $240.61 \pm 77.00$ |
| Ubuntu IRC | $1.00 \pm 0.00$ | $0.02 \pm 0.01$ | $0.00 \pm 0.00$ | $67.01 \pm 13.77$ |
| USPTO | $0.40 \pm 0.21$ | $0.02 \pm 0.01$ | $0.60 \pm 0.21$ | $1530.14 \pm 203.06$ |
| Wikipedia | $0.89 \pm 0.11$ | $0.02 \pm 0.01$ | $0.11 \pm 0.11$ | $621.42 \pm 211.98$ |
| YoutubeSubtitles | $0.97 \pm 0.06$ | $0.02 \pm 0.02$ | $0.03 \pm 0.06$ | $238.47 \pm 118.93$ |

We introduced the approximate Kelly betting strategy as an additional option for allocating bets within our sequential e-value testing framework in Appx. D.4. Our goal in this section is to demonstrate that the statistical power of the test is robust. Specifically, it does not depend sensitively on the choice of betting strategy nor on the maximum betting cap.

To quantify this robustness, we compute the *sequential power curve* across repeated Monte Carlo simulations. The sequential power at time $t$ is defined as the probability that the test has rejected the null hypothesis on or before time $t$. Formally, for a given $t$, let $T_i$ denote the stopping time for the $i$-th trial. The empirical sequential power is then estimated as:

$$\text{Power}(t) = \frac{1}{\text{num. trials}} \sum_{i=1}^{\text{num. trials}} \mathbf{1}\{T_i \leq t\}.$$

As shown in Figure 16, we plot the sequential power curves for the Ubuntu subset of the Pile dataset using MI features extracted by our method on the Pythia–12B model. The results demonstrate that the hypothesis test is robust to both the choice of betting strategy and the maximum betting cap. In accordance with log–optimality constraints, the maximum betting value must lie within the interval $[-1, 1]$ to prevent the wealth process from collapsing to zero. Within this feasible range, both the ONS and approximate Kelly strategies achieve their highest empirical power at a maximum betting value of $0.6$. However, the sequential power remains stable across different cap values, consistently converging to the significance threshold under the alternative hypothesis.

## D.6    ROBUSTNESS TO POLLUTED HELD-OUT SET (VALIDATION DATA)

To evaluate the robustness of our sequential testing procedure under held-out-set (validation data) contamination, we conduct an experiment in which a specified proportion of non–members is replaced with true training members. We then apply our hypothesis testing by betting method to the Ubuntu subset of the Pile dataset (Gao et al., 2020) using the Pythia–12B model (Biderman et al., 2023).

In this framework, the bettor adaptively places stakes based on all observations seen so far. Under the alternative hypothesis, accumulated evidence causes the wealth process to grow. We execute the experiment over 50 random seeds and report the average wealth trajectory in Figure 17. For pollution

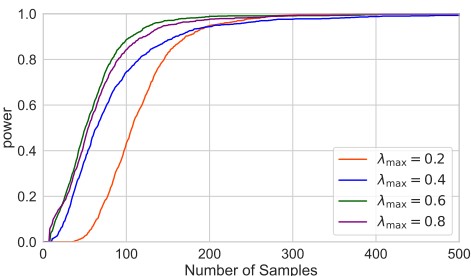 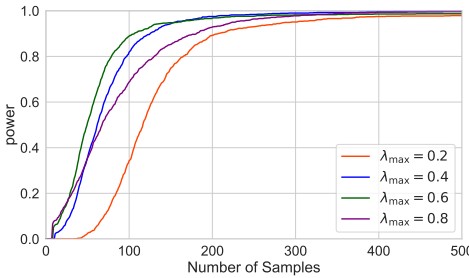

(a) Sequential Power Curve (Kelly Betting).   (b) Sequential Power Curve (ONS Betting).

Figure 16: **Robustness of Sequential Power Across Betting Strategies.** We compare sequential power curves for the approximate Kelly strategy and the ONS strategy on the Ubuntu subset. Results show that detection performance is consistent across strategies.

levels up to $70\%$, the test reliably gathers sufficient evidence to cross the $1\%$ significance threshold within the first 200 observations. At $80\%$ pollution, the rate of wealth growth diminishes, but the trajectory still shows consistent upward movement, indicating that the remaining clean examples provide enough evidence against the null. At $90\%$ pollution, the curves flatten substantially, reflecting that the contaminated distribution becomes nearly indistinguishable from the null.

Overall, the results demonstrate that the betting-based sequential test is robust to substantial contamination of the held-out set (validation data). Even with as much as $70\%$ of the non–members replaced by members, the procedure continues to accumulate evidence and successfully rejects the null hypothesis.

A notable aspect of Figure 17 is that the curve for a $30\%$ pollution ratio reaches the $1\%$ significance threshold with fewer observations than the $20\%$ curve, and even performs comparably to the clean ($0\%$) baseline. At first glance, this appears counter–intuitive, since higher contamination should make the hypothesis test harder. However, this behavior is fully consistent with the properties of the wealth process.

The e–value process is a stochastic, path–dependent quantity. In our experiments, each curve represents an average over 50 random seeds, where the order of samples is independently shuffled at each run. Because the sequential test operates online, observing data points one at a time, early observations with a large discrepancy between the null and alternative can yield disproportionately large multiplicative jumps in the wealth. Consequently, contamination can occasionally introduce early data points that appear highly informative to the bettor, causing faster boundary crossing even when the overall setting is less favorable.

Importantly, theory does not require that a "stronger" or cleaner alternative always leads to earlier stopping. For e–processes, the only guarantee is that under the alternative the wealth will eventually grow and cross the threshold with high probability; it does not impose monotonicity of stopping times across different alternatives or contamination levels. The variability observed in Figure 17 therefore reflects inherent stochasticity in the sequential evidence accumulation process, rather than a violation of the expected theoretical behavior.

### D.7 Robustness to non-member selection

To further evaluate the robustness of the hypothesis testing by betting framework with respect to the selection of non-member data points, we used the MI scores extracted by our method from the Pythia-12B model on the Ubuntu subset of The Pile dataset. We randomly shuffled the data and divided it into batches of 100 data points each. We then applied the sequential test independently to every batch. The full dataset contains 2,000 member and non-member data points, resulting in 20 subsets of 100 points each. For every subset, we ran the test with 50 random seeds and plotted the average log wealth across seeds.

The results are shown in Figure 18. In every batch, the test successfully accumulates sufficient evidence to reject the null hypothesis and reaches the $1\%$ significance threshold. This demonstrates that the test is robust to the choice of non-member samples: as long as the discrepancy metric provides

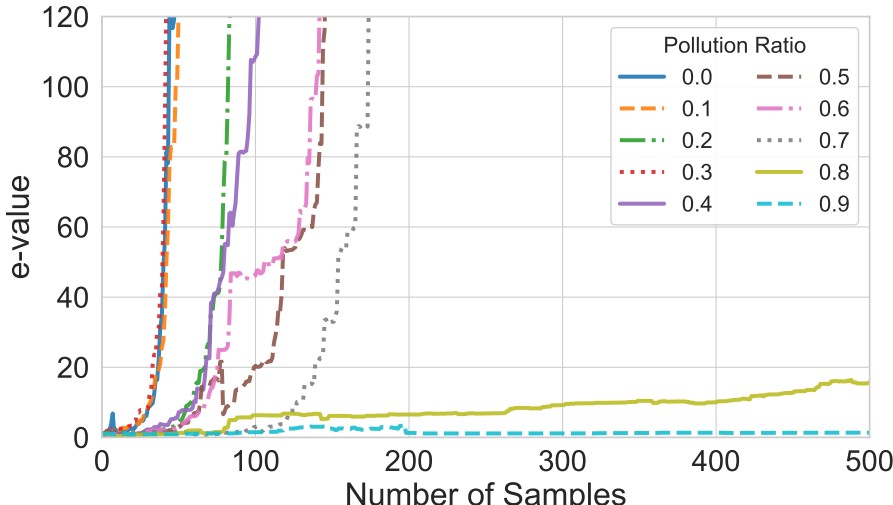

Figure 17: **Effect of Pollution Ratio in the Held-Out Set on Sequential e-value Growth.** The plot shows the mean wealth (e-value) trajectories for different non-member pollution ratios ranging from 0 to 0.9. Each curve corresponds to a different injected pollution level. The curves show that once sufficient evidence accumulates against the null hypothesis, the wealth process increases and eventually crosses the significance threshold.

enough evidence, the betting strategy allocates wagers so that the wealth grows exponentially under the alternative.

# E    GENERALIZATION TO DIFFERENT MODEL FAMILIES AND ACCESS LEVELS

**OLMo-7B.** We apply the BADI framework to the OLMo-7B model by conducting sequential hypothesis testing via betting. Our evaluation is performed on multiple subsets of the Dolma dataset (Soldaini et al., 2024), and the results are shown in Figure 19. Across diverse data sources (including Reddit, StackExchange, Common Crawl, Wikipedia, Gutenberg, and PeS2o) BADI consistently detects membership, with e-values surpassing 100 for most subsets. Although the Reddit subset does not yet exceed the 5% significance threshold, the e-value exhibits a steadily increasing trend as more data are observed, indicating that the test is on track toward rejection. These findings demonstrate that BADI remains reliable and effective across heterogeneous data distributions and model families. Additionally we compare BADI with PETAL and Baseline method using the AUC metric. The results are reported in Table 10

**GPT-3.5.** We further evaluate our DI method in a black-box setting against the commercial GPT-3.5 API using the BookMIA 2023 dataset `https://huggingface.co/datasets/swj0419/BookMIA`. As shown in Figure 20, our approach consistently succeeds in this restricted setting: it crosses the 1% significance threshold in fewer than 200 samples and maintains uniform FPR control across 1,000 observations. We conduct sequential testing by betting with 50 independent random seeds. On average, our method achieves a TPR of 0.995 with an average stopping time of 125.04 observations.

**Qwen2-7B.** To further evaluate the robustness of BADI across different model families, we assess our method on both the QWEN2-7B BASE and QWEN2-7B-INSTRUCT (Yang et al., 2024) models using the BOOKMIA 2023 dataset. The results in Figure 21 demonstrate a significant effect in the wealth process, indicating that BOOKMIA data was indeed included in the training set of QWEN2-7B.

Moreover, our method not only successfully infers membership for the BOOKMIA dataset, but also remains effective for instruction-tuned models designed for improved alignment. As shown in Figure 21, the instruction-tuned model requires slightly more observations to reach the 1%

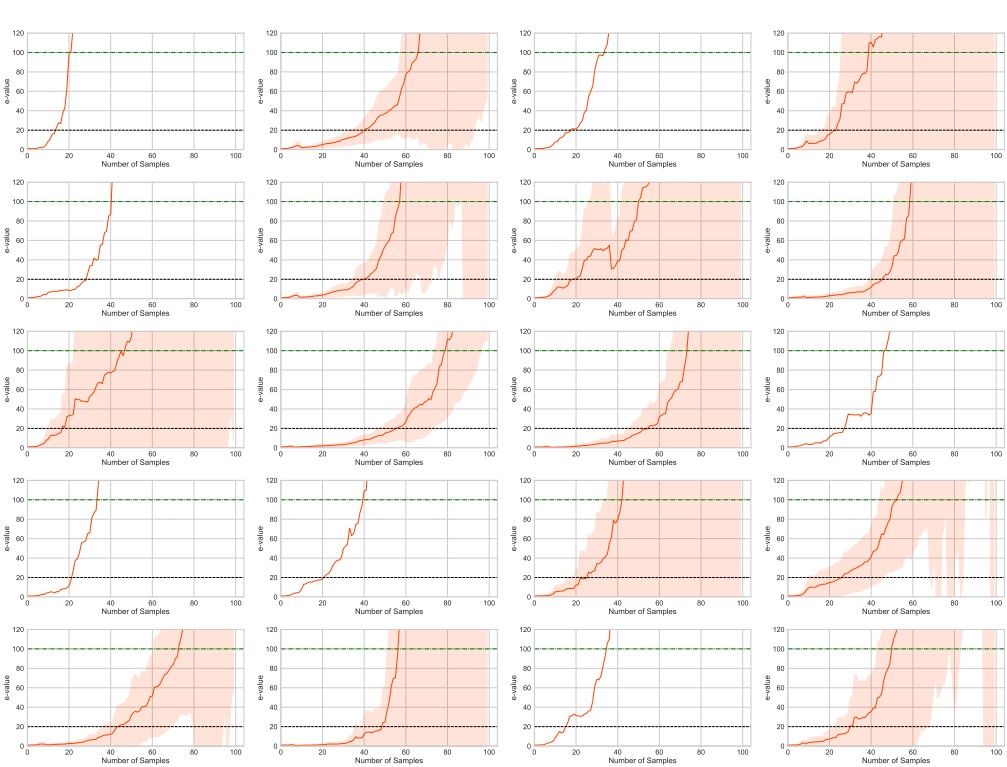

Figure 18: **E-process trajectories for 20 batches of 100 data points from the Ubuntu subset of the Pile dataset**. We use MI features collected with our method from the Pythia-12B model. The results show that the betting-based testing framework is robust in distinguishing non-member data points, consistently accumulating sufficient evidence to reject the null hypothesis in each subset independently.

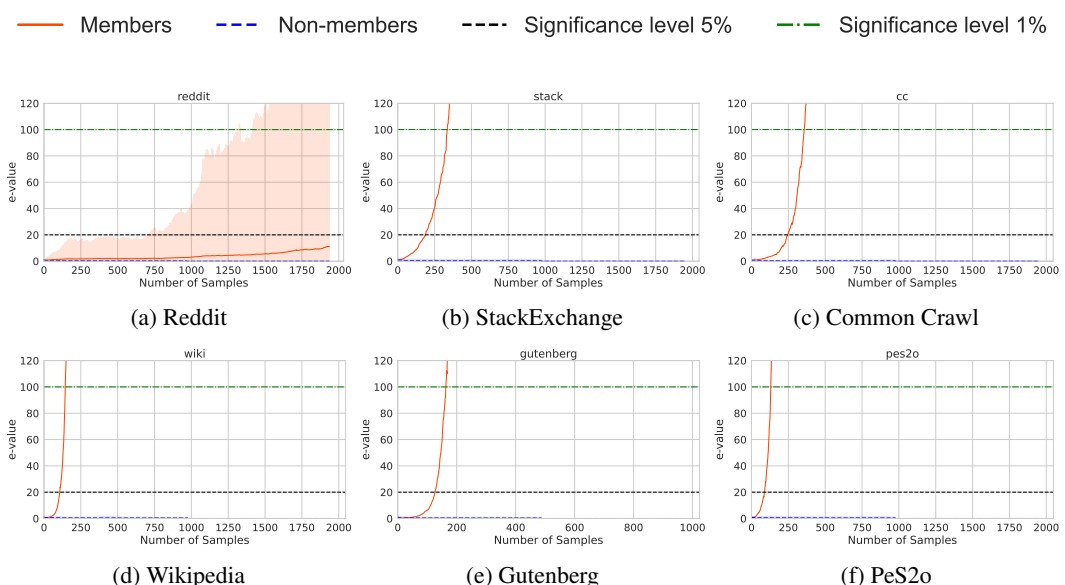

Figure 19: **Our Testing by Betting.** We present accumulated wealth with 95% confidence intervals of the e-process performed on various subsets on the OLMo-7B model.

Table 10: **AUC results of DI on Dolma dataset and OLMo-7B.** We compare our *BADI* with PETAL and Baseline methods.

| Dataset | Baseline | PETAL | BADI (Ours) |
|---|---|---|---|
| Gutenberg | 0.664 | 0.586 | **0.801** |
| Common Crawl | 0.531 | **0.619** | 0.612 |
| PeS2o | 0.446 | 0.451 | **0.762** |
| Reddit | 0.482 | 0.432 | **0.555** |
| StackExchange | 0.501 | 0.532 | **0.738** |
| Wikipedia | 0.484 | 0.544 | **0.762** |

significance threshold. We hypothesize that this additional difficulty may be due to perturbations introduced into the model's output distribution during the instruction-tuning process.

## F  LLM USAGE DECLARATION

We used large language models for refining text, focusing exclusively on style, grammar, and spelling, without compromising the original semantic meaning.

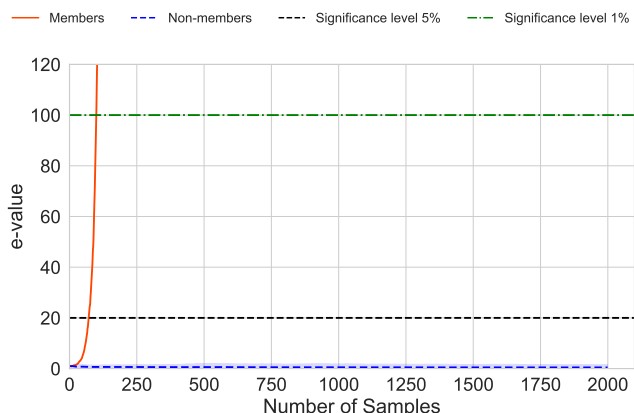

Figure 20: **BADI for GPT-3.5.** Accumulated wealth trajectory for the BookMIA 2023 dataset with GPT-3.5 as the target model.

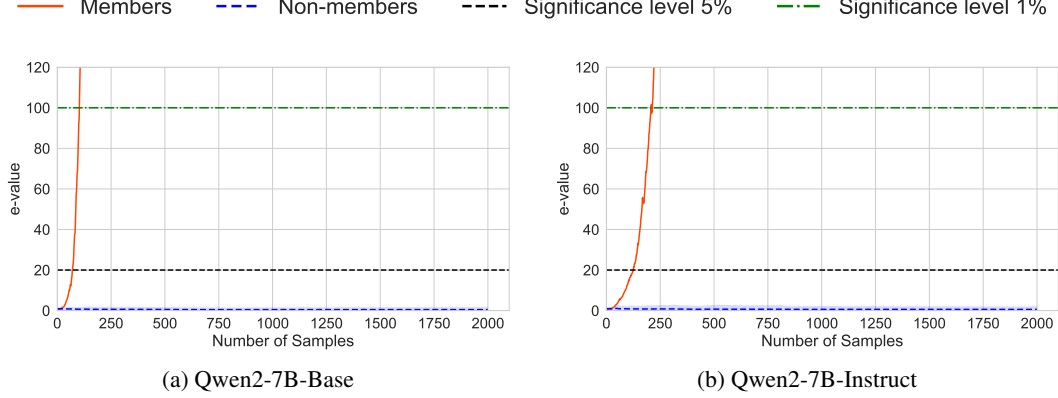

(a) Qwen2-7B-Base                    (b) Qwen2-7B-Instruct

Figure 21: **BADI for Qwen2 model variants.** The results show that BADI successfully detects the membership of the BOOKMIA dataset in both models. In addition, the instruction-tuned model requires slightly more observations to reach the $1\%$ significance threshold, suggesting that instruction tuning may introduce perturbations to the output distribution that make membership detection more challenging.

