# OpenReview forum: "BADI: Black-box and Anytime-valid Dataset Identification for Large Language Models"
_ICLR.cc/2026/Conference — Submitted to ICLR 2026_

### Official Review · Reviewer_ck6T · 2025-10-29

**Soundness:** 3
**Presentation:** 3
**Contribution:** 2
**Rating:** 4
**Confidence:** 3

**Summary:**

This paper introduces BADI, a black box method for dataset identification. The proposed black-box DI framework used e-values for hypothesis testing.

**Strengths:**

- This paper is well-organized.
- As the authors stated, the testing framework is based on e-values. This could introduce more flexibility in the task.

**Weaknesses:**

- Only the Pythia model family is studied in this work.
- The rationale of using Eq.(1) for token probabilities needs further justification.
- Few baselines are studied in the experiment section.

**Questions:**

- In the experiment stage, only the Pythia model family is studied. This is a model released at 04.2023, it is trained on **300B tokens** [1]. Typically, today, LLMs like Meta-Llama-3.1 [2], "We pre-train Llama 3 on a corpus of about **15T multilingual tokens**, compared to 1.8T tokens for Llama 2." The Pythia model only consumes **1/50 tokens** of Meta-Llama-3.1. And the training tokens are always increasing, for example, "Qwen3 has been pre-trained on **36 trillion tokens**"[3].

- With the increasing of training tokens, many questions occur. For example, there is a large chance that similar examples to the test one exist, and this will lead to false-positive results. It is a well-known problem of the traditional DI/MIA method [4]. When nearly all web data is used for training, the detection will become hard, and choosing/creating a held-out set is a problem.

- A summary of the above two points, the reviewer believes that we should conduct more analysis with recent models rather than only using the Pythia model. This could give us more information about the proposed method. Can it really deal with modern LLMs training on web-scale data?

- For Eq.(1), it assumes a token-to-token mapping relationship between the generated sentence and ground truth. Will a semantically similar generated sentence get a low score with the ground truth? It may have different word orders, but basically the same meaning. How do we deal with and consider this case?

- In Figure 3, it seems that the reviewer only provides PETAL and Baseline for comparison. How about the performance of other DI/MIA methods? Could the authors provide other metrics like AUC?

[1] https://huggingface.co/EleutherAI/pythia-12b

[2] https://arxiv.org/pdf/2407.21783

[3] https://arxiv.org/pdf/2505.09388

[4] Do Membership Inference Attacks Work on Large Language Models? https://arxiv.org/pdf/2402.07841

---

> ### Author Response · Authors · 2025-11-24
> **Responses to Reviewer ck6T (1/2)**
>
> We thank the Reviewer for their insightful comments and for recognizing our framework as more flexible and our paper as well organized.
>
> Below, we address each of the points one-by-one in detail.
>
> >**W1: Only the Pythia model family is studied in this work.**
> >**Q1:  The Reviewer believes that we should conduct more analysis with recent models rather than only using the Pythia model. This could give us more information about the proposed method. Can it really deal with modern LLMs training on web-scale data?**
>
> We thank the Reviewer for this valuable perspective, which has helped us further strengthen the scope of our experimental evaluation. Our initial choice to evaluate the Pythia model suite follows the established convention in prior dataset inference work [1,2,3,4], allowing for direct and fair comparison with existing approaches. To expand beyond this family of unaligned pretrained models, we have additionally incorporated experiments on the OLMo-7B model. The results on the different subsets of Dolma dataset successfully passed the significance threshold of 1% detecting the membership. The results are presented in Appendix **E** in the updated version of our submission.
>
> In line with the Reviewer’s suggestion to assess performance on large-scale, internet-trained LLMs, we further evaluated our method on **GPT-3.5-Turbo** using the **BookMIA** dataset. GPT-3.5 Turbo model can understand and generate natural language or code and has been optimized for chat using the Chat Completions API but works well for non-chat tasks as well.
> We present the results in Figure 20 in **Appendix E** (in the updated version of our submission). Our approach consistently succeeds in this restricted setting with only the black-box access to the target model: it crosses the 1% significance threshold in fewer than 200 samples and maintains uniform FPR control across 1,000 observations. We conduct sequential testing by betting with 50 independent random seeds. On average, our method achieves a TPR of 0.995 with an average stopping time of 125.04 observations. This confirms that BADI remains effective not only on larger, more recent architectures, but also in a fully black-box commercial API setting, where access to model internals is restricted.
>
> >**W2: The rationale of using Eq.(1) for token probabilities needs further justification.**
>
> We introduce a novel *sigmoid-based calibration method* that calibrates token probability estimates without relying on a surrogate model. Specifically, we obtain per-token embeddings using a sentence embedding encoder. For each next token, we compute the semantic similarity (using the dot product) between the embedding of the predicted next token from the target model and that of the corresponding ground truth token. This similarity score is then transformed into a probability using the sigmoid-based function Eq.(1). We added the above text to the main part of the paper in Section 3 (blue font in the updated version of our submission). Additionally, we ablated the hyper-parameters in Eq.(1). We present the distribution of p-values for different scaling factors of the sigmoid function across all Pile subsets on Pythia-12B-dedup in Figure 6 in the Appendix A.6. We evaluate scaling factors 0.5, 2, 10 and observe that the factor 2 yields the strongest separation between members and non-members.
>
> >**Q2: For Eq.(1), it assumes a token-to-token mapping relationship between the generated sentence and ground truth. Will a semantically similar generated sentence get a low score with the ground truth? It may have different word orders, but basically the same meaning. How do we deal with and consider this case?**
>
> This question is related to the above comment about the rationale behind the Eq.(1). Since we measure the token similarities between the generated vs ground-truth tokens, the more the generated tokens are semantically similar to the ground-truth tokens, the higher the score. We could further consider a larger window than a single token for the semantic similarity, however, we observed that the per-token similarities already provided us with the perfect performance on the Pile dataset using the Pythia suite of models and further on the additionally tested OLMo 7B and GPT-3.5-Turbo models.

---

> ### Author Response · Authors · 2025-11-24
> **Responses to Reviewer ck6T (2/2)**
>
> >**W3: Few baselines are studied in the experiment section.**
> >**Q3: In Figure 3, it seems that the Reviewer only provides PETAL and Baseline for comparison. How about the performance of other DI/MIA methods? Could the authors provide other metrics like AUC?**
>
> To further strengthen our empirical analysis, we have also included ROC curves (Fig. 14) and AUC scores (Table 1, Table 2, Table 3) comparing our method against PETAL and the Baseline. These results demonstrate a clear performance advantage for our approach, indicating a substantial improvement over existing methods.
>
>
> **References:**
>
> [1] Maini et al. “*LLM Dataset Inference: Did you train on my dataset?*” NeurIPS 2024.
>
> [2] Zhao et al. “*Unlocking Post-hoc Dataset Inference with Synthetic Data*” ICML 2025.
>
> [3] Rastogi et al. “*STAMP Your Content: Proving Dataset Membership via Watermarked Rephrasings*” ICML 2025.
>
> [4] Rossi et al. “*Privacy Auditing for Large Language Models with Natural Identifiers*” ICLR 2025 Workshop on Navigating and Addressing Data Problems for Foundation Models.
>
> ---
>
> We thank the Reviewer for the valuable questions, which help make our submission more solid and clear. If our rebuttal addresses the Reviewer's concerns, we would appreciate it very much if  they consider increasing the rating. We are also happy to address any remaining questions.

---

> ### Comment · Reviewer_ck6T · 2025-11-27
> **Response**
>
> Hi Authors,
>
>
> Thanks for your responses. They addressed most of my concerns. I think this is an interesting work, but it will still need more verifications from more popular LLMs.
>
> I will raise my score to 6.
>
> Thank You.
>
> Best.

---

> > ### Author Response · Authors · 2025-11-28
> > **Additional Experiments - BADI applied to Qwen2 models**
> >
> > We thank the reviewer for finding our work interesting and for raising our score. We appreciate the suggestion regarding validation on more popular LLMs. In addition to our results on GPT-3.5 (Fig. 20), we have now included experiments on both **Qwen2-7B-Base** and **Qwen2-7B-Instruct** models using the **BookMIA** dataset (Fig. 21) in **Appendix E**. The accumulated wealth trajectories indicate a statistically significant effect under our betting-based hypothesis testing framework, confirming that **BADI successfully detects membership in both variants of Qwen2**.
> >
> > Furthermore, these results highlight an additional important point: **BADI remains effective even for instruction-tuned models**, where alignment-related distribution perturbations could potentially make detection more difficult. Combined with our results on **GPT-3.5** (showing applicability to commercial black-box APIs) and **OLMo-7B** (a fully open model), our experiments now demonstrate that **BADI generalizes across multiple families of models with different levels of alignment and accessibility**.
> >
> > We appreciate the reviewer’s valuable feedback, which helped strengthen our experimental validation. If the newly added results sufficiently address your concern, we would be grateful if you would consider adjusting the score accordingly. We are happy to provide any further clarification or conduct additional experiments if needed.

---

### Official Review · Reviewer_Xq4R · 2025-10-31

**Soundness:** 2
**Presentation:** 3
**Contribution:** 2
**Rating:** 4
**Confidence:** 3

**Summary:**

This paper proposes BADI, a new framework for black-box and anytime-valid dataset inference in LLMs. The goal is to determine whether a suspect dataset was used during an LLM’s pretraining, addressing concerns around copyright and data transparency. The method is empirically evaluated on multiple Pythia models (410M, 6.9B, 12B) and diverse Pile subsets. Experiments show that BADI identifies training datasets with higher efficiency (fewer samples required) and lower false-positive rates than baselines (PETAL and a RoBERTa-similarity baseline), offering a practical framework for real-world LLM auditing.

**Strengths:**

1. The paper situates itself well within the context of data auditing, copyright compliance, and LLM transparency.
2. The authors argued that they proposed the first black-box DI framework that avoids reliance on per-token logits.
3. The step-by-step framework (Figure 1) is clear and pedagogically strong.

**Weaknesses:**

- The paper claims generality to “API-based black-box models,” but does not empirically test BADI on true commercial APIs (GPT, Claude, Gemini). All experiments are performed on open Pythia models (Biderman et al., 2023). A key claim "BADI enables accurate dataset identification in black-box settings" is not fully validated since Pythia allows full access.
- While e-values provide anytime-validity, the choice of betting strategy (λt bounds, ONS parameters) may significantly affect power. No experiments systematically compare BADI’s sequential testing to classical p-value–based tests under equivalent conditions.
- Some formulae (e.g., kernel MMD payoff mapping) and hyperparameters for the online regressor are only briefly described, lacking mathematical detail or convergence justification. Morover, the description of “STRIP-K% PROB” lacks sensitivity analysis across K values.

**Questions:**

- Have the authors tested BADI on real black-box APIs (e.g., OpenAI, Anthropic, Google)? If not, how confident are they that the token similarity estimation remains meaningful when model outputs are diverse or truncated?
- How was the sigmoid mapping in Eq. (1) empirically chosen? Is there a theoretical justification for using 2·sim as the scaling factor, and how sensitive is performance to this parameter?
- The paper adopts an Online Newton Step (ONS) for adaptive staking. How critical is this choice? Would simpler strategies (fixed λ or proportional betting) degrade performance substantially?
- The paper reports ≈1 % false positive at 5 % significance. Was this averaged over datasets, or verified under a global Type-I control across all tests?

---

> ### Author Response · Authors · 2025-11-24
> **Responses to Reviewer Xq4R (1/3)**
>
> We thank the Reviewer for the constructive feedback and for recognizing our framework as new, clear, and strong, and the whole paper as well positioned.
>
> We address each of the comments and questions in detail one-by-one below.
>
> >**W1: The paper claims generality to “API-based black-box models,” but does not empirically test BADI on true commercial APIs.** & **Q1: Have the authors tested BADI on real black-box APIs (e.g., OpenAI, Anthropic, Google)? If not, how confident are they that the token similarity estimation remains meaningful when model outputs are diverse or truncated?**
>
> We thank the Reviewer for this valuable perspective, which has helped us further strengthen the scope of our experimental evaluation. Our initial choice to evaluate the Pythia model suite follows the established convention in prior dataset inference work [1,2,3,4], allowing for direct and fair comparison with existing approaches. To expand beyond this family of pretrained models, we have additionally incorporated experiments on the OLMo-7B model and GPT-3.5 Turbo.  For OLMo-7B, the results on the different subsets of Dolma dataset successfully passed the significance threshold of 1% detecting the membership.
>
> In line with the Reviewer’s suggestion to assess performance on large-scale, internet-trained LLMs, we further evaluated our method on **GPT-3.5-Turbo** using the **BookMIA** dataset. GPT-3.5 Turbo model can understand and generate natural language or code and has been optimized for chat using the Chat Completions API but works well for non-chat tasks as well. Our results demonstrate that our method for sequential testing over 50 random seeds (Figure 20 in the updated submission) crosses the 1% significance threshold. This confirms that BADI remains effective not only on larger, more recent architectures, but also in a fully black-box commercial API setting, where access to model internals is restricted.
>
> Our full results are presented in Appendix **E** in the updated version of our submission.
>
> >**Q2: How was the sigmoid mapping in Eq. (1) empirically chosen? Is there a theoretical justification for using 2·sim as the scaling factor, and how sensitive is performance to this parameter?**
>
> The scaling factor in the sigmoid mapping of Eq. (1) was selected based on both empirical evaluation and practical considerations. We evaluated three approaches for translating similarity scores into probability estimates: (1) raw similarity values, (2) reference-model calibration (as in PETAL), and (3) a sigmoid-based mapping (our method). As shown in our original submission (Appendix A.6, Fig. 7), the sigmoid mapping consistently produced more accurate probability estimates.
>
> Regarding the choice of the scaling factor, we tested multiple values {0.5, 2, 10} via grid search. The factor of **2** yielded the strongest separation between members and non-members and achieved the best downstream hypothesis-testing performance. This scaling expands the effective dynamic range of the estimated probabilities without saturating the sigmoid too early, which is important for capturing subtle differences in membership signals. Empirically, **2** was the most effective and robust choice across datasets.
>
> >**W2: Some formulae (e.g., kernel MMD payoff mapping) and hyperparameters for the online regressor are only briefly described, lacking mathematical detail or convergence justification. Moreover, the description of “STRIP-K% PROB” lacks sensitivity analysis across K values.**
>
> The hypothesis-testing stage of BADI involves two primary hyperparameters: (1) the number of Monte-Carlo repetitions, and (2) the maximum betting threshold $\lambda_{max}$.
>
> **1. Monte-Carlo repetitions.**
>
> This parameter specifies how many independent trials are run for a fixed sample size and distributional setup. Since type-I error and statistical power are probabilities, they cannot be estimated from a single execution of the test; repeated trials are required to empirically approximate these quantities. Increasing the number of repetitions decreases variance in the empirical estimates and yields smoother, more reliable power and error curves. It does **not** change the behavior of the test, it only improves the precision of the evaluation.

---

> ### Author Response · Authors · 2025-11-24
> **Responses to Reviewer Xq4r (2/3)**
>
> **2. Maximum betting threshold ($\lambda_{\max}$).**
>
> In BADI, $\lambda_{\max}$ restricts the aggressiveness of the sequential betting process used to build the e-value. A higher $\lambda_{\max}$ accelerates e-value growth under the alternative hypothesis but risks over-betting; a smaller $\lambda_{\max}$ yields more conservative but safely valid behavior. Thus, $\lambda_{\max}$ controls the tradeoff between statistical power and guaranteed type-I validity. We choose $\lambda_{\max}$ within recommended bounds ensuring non-parametric, anytime-valid hypothesis testing. An ablation on $\lambda_{\max}$ and its effect on test power is provided in **Appendix D5**, along with an extended discussion.
>
> **Hyperparameters of the Online Regressor.**
>
> The neural network used in the online regressor is **not a statistical component whose convergence guarantees validity**. It serves only as a scoring function to define the payoff sequence in the betting procedure. Validity (anytime type-I control) follows from the e-value martingale construction and **remains correct even if the regressor has not fully converged**. Training quality influences *power*, not *validity*.
>
> In practice, we use a logistic model with binary cross-entropy loss and Adam optimizer, trained incrementally using accumulated samples. The choice of **30 epochs per update** is purely empirical: loss stabilization occurs well before 30 epochs, and further training provides negligible improvement to the score sequence.
>
> **Use of Multiple K-STRIP Thresholds.**
>
> BADI does **not** rely on selecting a single STRIP threshold K. Instead, we compute STRIP features over multiple thresholds (10%–40%) and jointly feed them into a linear scoring model within the betting framework. This design yields automatic sensitivity adaptation: the model learns appropriate weights across thresholds, up-weighting informative ones and down-weighting those with weaker discriminative value. In other words, robustness is achieved through aggregation, not tuning a single K value. The learned weight patterns across datasets are shown in **Figure 8 of the main submission**, demonstrating consistent stability across subsets of the Pile dataset.
>
> > **Q3: The paper adopts an Online Newton Step (ONS) for adaptive staking. How critical is this choice? Would simpler strategies (fixed λ or proportional betting) degrade performance substantially?**
>
> The choice of using **Online Newton Step (ONS)** is motivated by the objective of sequential e-value testing: the ideal betting fraction is the value that maximizes the **expected log-growth of wealth under the alternative hypothesis**. In theory, the optimal fraction is the solution to
>
> $$
> \lambda^* = \arg\max_{\lambda \in [-1,1]} \ \mathbb{E}_{\mathbb{P}_1}[\log(1 + \lambda \cdot Z)],
> $$
>
> where $Z$ denotes the observed payoff variable derived from the witness function. However, $\lambda^*$ is unknown because it depends on the (unknown) underlying effect size and witness distribution. Therefore, a **data-adaptive betting strategy** is required to approximate this optimal choice while maintaining stability.
>
> ONS satisfies this requirement because it provides a **provably bounded regret guarantee** (Appendix C.1, with proof adapted from [5]). This implies that ONS asymptotically tracks the best constant betting fraction **without risking catastrophic wealth collapse under the null**, and without being overly conservative under the alternative. In contrast:
>
> - **Fixed-$\lambda$** betting performs well *only* if the preselected $\lambda$ coincides with the true effect size.
>   - If $\lambda$ is too large ⇒ wealth can collapse to zero.
>   - If $\lambda$ is too small ⇒ the test accumulates evidence too slowly, reducing power.
>
> - **Naïve (proportional/Kelly-style) betting** lacks regret control and can behave poorly when early empirical signals fluctuate, especially when payoff estimates are noisy.
>
> **Additional Experiments: Robustness to Betting Strategy.**
>
> To directly address the Reviewer’s concern, we additionally evaluated a **simplified approximate-Kelly betting strategy**, similar to that used in [6], while enforcing theoretical safety bounds that guarantee nonnegativity and log-optimality. **Table 9** reports the TPR, FPR, and FNR for this strategy on **Pythia-12B** using our MI-based features. The results confirm that as long as these theoretical constraints are respected, the sequential test reliably crosses the significance threshold under the alternative.

---

> ### Author Response · Authors · 2025-11-24
> **Responses to Reviewer Xq4R (3/3)**
>
> **Effect of Maximum Betting Threshold.**
>
> To further analyze robustness, we varied the **maximum betting fraction**
>
> $$
> \lambda_{\max} \in \{0.2, 0.4, 0.6, 0.8\}
> $$
>
> for both ONS and approximate Kelly. For each setting, we computed the **sequential power curve** by estimating the probability that the hypothesis test rejects on or before time $t$. Formally, for $N$ Monte Carlo trials:
>
> $$
> \text{Power}(t) = \frac{1}{N} \sum_{i=1}^{N} \mathbf{1}\{T_i \le t\}
> $$
>
> where $T_i$ denotes the stopping time of the $i$-th trial.
>
> **Figure 13** shows that the test remains highly robust to both $\lambda_{\max}$ and the choice between ONS and approximate Kelly, with only minor variation in early-time evidence accumulation.
>
> > **Q4: The paper reports ≈1 % false positives at 5 % significance. Was this averaged over datasets, or verified under a global Type-I control across all tests?**
>
> All reported false positive rates are computed separately for each model and each subset of the PILE dataset, as shown in Tables 4–9. For every such configuration, we run the full sequential test pipeline using 50 different random seeds, and the final TPR/FPR/FNR values reported in the tables are the averages over these seeds.
>
> For each seed, the TPR, FPR, and FNR values are obtained empirically from Monte Carlo simulations of the sequential test under both the alternative and null setups. Specifically, for a given configuration we run 1,000 independent Monte Carlo trials of the sequential test:
>
> Under the alternative hypothesis, a true positive is counted whenever the test rejects (i.e., the wealth process crosses the $(1/\alpha)$ threshold with $\alpha=0.05$). Under the null hypothesis, a false positive is counted whenever a rejection occurs despite the two distributions being constructed to be indistinguishable. Trials with no rejection contribute to false negatives and true negatives, respectively. From these empirical counts, we compute TPR, FPR and FNR.
>
> This procedure is repeated for all 50 seeds, and the reported FPR values are the averages over these repetitions, performed independently for each dataset subset, not pooled across datasets.
>
> Regarding Type-I error control, the sequential e-value test guarantees that, under the null, the wealth process is a nonnegative supermartingale with $\mathbb{E}[W_t] \le 1$. In our experiments, this guarantee is reflected empirically: in the false-positive simulations (nonmember vs. nonmember), the wealth process nearly always remains below the rejection threshold, resulting in an observed FPR consistently around 1%, well below the nominal 5% significance level. This confirms that the method maintains Type-I error control in practice for every dataset subset considered.
>
> **References:**
>
> [1] “*LLM Dataset Inference: Did you train on my dataset?*” Maini et al. NeurIPS 2024.
>
> [2] “*Unlocking Post-hoc Dataset Inference with Synthetic Data*” Zhao et al. ICML 2025.
>
> [3] “*STAMP Your Content: Proving Dataset Membership via Watermarked Rephrasings*” Rastogi et al. ICML 2025.
>
> [4] “*Privacy Auditing for Large Language Models with Natural Identifiers*” Rossi et al. ICLR 2025 Workshop on Navigating and Addressing Data Problems for Foundation Models.
>
> [5] *"Black-Box Reductions for Parameter-free Online Learning in Banach Spaces"* Ashok Cutkosky and Francesco Orabona. 31st Annual Conference on Learning Theory 2018.
>
> [6] *"Non-parametric two sample testing by betting"* Shubhanshu Shekhar and Aaditya Ramdas. IEEE Transactions on Information Theory 2023.
>
> ---
>
> We thank the Reviewer for the valuable questions, which help make our submission more solid and clear. If our rebuttal addresses the Reviewer's comments, we would appreciate it very much if they consider increasing the rating. We are also happy to address any remaining questions.

---

### Official Review · Reviewer_ZAf3 · 2025-10-31

**Soundness:** 3
**Presentation:** 3
**Contribution:** 3
**Rating:** 4
**Confidence:** 3

**Summary:**

This paper presents a method for identifying if certain training data has been used in the training of an LLM. They do this also with black box models behind an API by approximating per token probabilities.

The estimation of token probabilities for black box models is done with a surrogate model and a computation of semantic similarity between the surrogate model's response and the true token predicted by the black box model. These are mapped to token probabilities.

**Strengths:**

The paper presents a novel approach for both estimating token probabilities in black box settings as well as predicting inference attacks

**Weaknesses:**

Existing LLMs, when they are properly aligned, they are not really trained to generate continuations but to answer with reasoning given a query. In such setting, any inference membership extraction method will be limited. The evaluation is done with the Pythia models which are trained on subsets of the Pile dataset. These models are not aligned by default so the applicability of the method will be limited about this. Please let me know if I misinterpreted this.

This is a small concern, but I would suggest to define e-values before describing why they are suited to this particular issue or at least refer to the section where this is mentioned.

**Questions:**

Have you tested the estimation of token probabilities with models that you have access to the actual token probabilities?

Would this approach work with models that are fully aligned with RL optimization or similar methods?

---

> ### Author Response · Authors · 2025-11-24
> **Responses to Reviewer ZAf3**
>
> We thank the Reviewer for the positive, encouraging, and constructive feedback. We appreciate the recognition of our contributions as novel in both (1) estimating token probabilities in black-box settings and (2) enabling dataset inference under this setting.
>
> Below, we address each of these comments and questions one-by-one in detail:
>
> >**W1: I would suggest to define e-values before describing why they are suited to this particular issue or at least refer to the section where this is mentioned.**
>
> We thank the Reviewer for their constructive suggestion aimed at improving the clarity of our presentation. In response, we have added a concise definition of *e-values* in the introduction (Lines 94–96), before discussing their relevance to our problem: *“An e-value is a non-negative measure of evidence against a null hypothesis whose expected value under the null does not exceed one.”* We also clearly reference Section 4, where a more detailed and formal treatment of e-values is provided. Additionally, we expanded the discussion of e-values and non-parametric testing by betting in Section 4 and Appendix C to further enhance clarity and completeness.
>
> >**Q1: Have you tested the estimation of token probabilities with models that you have access to the actual token probabilities?**
>
> We thank the Reviewer for this insightful question. The perfect estimation of token probabilities is not strictly necessary for our method. The critical factor is achieving sufficient separation between members and non-members, which directly improves dataset inference. We conducted the required experiments to verify this. The results are presented in Figure 6 and Figure 7 in the updated version of our submission. Furthermore, our method is compared to PETAL and Baseline regarding the wealth accumulation (in Figure 3) and according to this figure our method accumulates evidence against the null hypothesis with fewer observations than other approaches. This indicates that indeed our estimation method for log-probabilities is more successful than PETAL and Baseline for detecting membership.
>
> >**W2:  The evaluation is done with the Pythia models which are trained on subsets of the Pile dataset. These models are not aligned by default so the applicability of the method will be limited about this. & Q2: Existing LLMs, when they are properly aligned, they are not really trained to generate continuations but to answer with reasoning given a query. In such setting, any inference membership extraction method will be limited. The evaluation is done with the Pythia models which are trained on subsets of the Pile dataset. These models are not aligned by default so the applicability of the method will be limited about this. Please let me know if I misinterpreted this.**
>
> We thank the reviewer for the insightful question. We would like to provide the following experiments and clarifications regarding the question.
>
> The Pythia models have been the standard benchmarking choice for dataset inference in all prior works [1,2,3,4], and we follow this established setup to ensure comparability.
>
> **BADI works on RLHF-aligned production-level model.**
>
> To demonstrate applicability to RLHF-aligned, instruction-tuned LLMs, we evaluate BADI on the production-level black-box model GPT-3.5-Turbo using the BookMIA dataset. Although optimized for chat, GPT-3.5-Turbo performs well on non-chat tasks. As shown in Appendix E, our sequential tests over 50 random seeds (Fig. 20) cross the 1% significance threshold, confirming that BADI remains effective on modern architectures and in fully black-box commercial API settings.
>
> **More experiments on the pre-trained models**
>
> To broaden our evaluation beyond pretrained models, we ran experiments on OLMo-7B across multiple Dolma subsets (Reddit, Stack Exchange, Common Crawl, Wikipedia, Gutenberg, PeS2o), all of which successfully detected membership. We include these additional OLMo-7B and Dolma results in Appendix E. We also applied BADI to Qwen2-7B-Base and Qwen2-7B-Instruct on the BookMIA dataset; Figure 21 in Appendix E shows successful membership detection for both models.
>
> **References:**
>
> [1] “*LLM Dataset Inference: Did you train on my dataset?*” Maini et al. NeurIPS 2024.
>
> [2] “*Unlocking Post-hoc Dataset Inference with Synthetic Data*” Zhao et al. ICML 2025.
>
> [3] “*STAMP Your Content: Proving Dataset Membership via Watermarked Rephrasings*” Rastogi et al. ICML 2025.
>
> [4] “*Privacy Auditing for Large Language Models with Natural Identifiers*” Rossi et al. ICLR 2025 Workshop on Navigating and Addressing Data Problems for Foundation Models.
>
> —
>
> We hope that our responses and newly added experimental results satisfactorily address the Reviewer’s comments. We would appreciate a reconsideration of the evaluation should the clarifications and revisions resolve the issues raised.

---

> ### Author Response · Authors · 2025-11-28
> **Additional Experiments - Application of BADI on Instruction Tuned models**
>
> To further address the reviewer’s comments regarding the applicability of our method to aligned models, we conducted an additional experiment using the **Qwen2-7B-Base** and **Qwen2-7B-Instruct** models on the **BookMIA dataset**. Here, instruction tuning serves as another form of alignment, allowing us to further evaluate the robustness of our method across different model settings.
>
> The results of this experiment are presented in **Appendix E** and **Figure 21**. We observe that, for both the base and instruction-tuned models, our method successfully detects a significant effect, indicating that the BookMIA dataset was indeed used as part of the training data for the Qwen2-7B models. Notably, the instruction-tuned model required slightly more data points to reach the 1% significance threshold. We attribute this to instruction tuning slightly perturbing the model’s output probability distribution. Nevertheless, even under this shift, our method is still able to reliably detect membership significance.
>
> Taken together with our experiments on GPT-3.5, these results further demonstrate the ability of BADI to perform effectively on models with different forms of alignment.

---

> > ### Public Comment · ~Adam_Dziedzic1 · 2026-04-03
> > **Rebuttal**
> >
> > Dear Reviewer,
> >
> > Thank you very much for your positive feedback. We really appreciate the time and effort took to review our submissions. We understand the heavy workload reviewers face, and we are grateful for your consideration. If possible, we would greatly appreciate it if you could also include the positive comments you shared here in your review of our current submission.
> >
> > With kind regards,
> >
> > Authors

---

### Official Review · Reviewer_roVQ · 2025-10-31

**Soundness:** 3
**Presentation:** 3
**Contribution:** 3
**Rating:** 6
**Confidence:** 3

**Summary:**

Overall, this is a good practical work that addresses important issues in dataset inference. While the evaluation could benefit from additional robustness tests, I believe readers of ICLR would find this work useful and interesting. Hence, I believe it should be accepted.


The authors outline current limitations of dataset inference (DI) methods: they require gray-box access to token probabilities, and they rely on p-value statistical tests which incur high computational costs and allow “gaming” significance levels. The authors propose to estimate token probabilities and use “e-values” with sequential testing to overcome these limitations, leading to BADI. This is more practical, as specifying testing dataset size becomes a flexible, sequential process, and one does not require gray-box access to LLMs.

The general method is to aggregate dataset inference features (largely results of membership inference techniques) for a given dataset, construct an e-value from the features. The features often leverage token-level probabilities – these are estimated using a sigmoid-based calibration method which avoids use of a surrogate model. The semantic similarity of a produced token with a reference token is used as a proxy for the probability of the original token.  Membership inference features are computed from the estimated token probabilities. The membership features are then used to continuously train a scoring model. The scores are passed to a betting model which should accumulate evidence in the case of true positives (true training samples). One may stop the process if sufficient evidence has been gathered to reject the null hypothesis (that the samples were not part of training).

Experiments with Pythia+Pile validate the method presented by the authors. They compare BADI with two baseline black-box methods leveraging RoBERTa scores and PETAL. The experiments would benefit from additional robustness tests (see weaknesses).


Minor comments:
Line 131 consider rephrasing “However, most existing MI methods for LLMs, even the stronger gray-box attacks do not perform better than random guessing.” Perhaps “However, for most existing MI methods for LLMs, even the stronger gray-box attacks do not perform better than random guessing.”
Line 152: grammar issue “Consider a randomly ordered points”
Line 160 “BADI rely” -> “BADI relies”

**Strengths:**

The paper is strongly motivated in practical scenarios and presents a useful solution to a challenging problem: dataset inference in black-box scenarios with flexible resource expenditure

The methods of the paper are sound, and the claims are largely substantiated by the experiments with Pythia+Pile.

**Weaknesses:**

The description of estimating token probabilities is not very clear. How is this semantic similarity measured? The “ground truth sequence” appears to be the one you would like to estimate the probability for – it would be helpful to refer to it as that if so.

There is not a study of robustness to choice of held-out distribution data. What if your trusted hold-out data is polluted with training samples (some percentage) or its distribution does not match that of the training samples? How does this affect evidence growth rate?

I would like to see more comparisons with additional dataset inference (DI) methods, however I believe there could be difficulties in creating apples-to-apples comparisons between p-value methods and e-value methods.

**Questions:**

How sensitive is the method to choice of non-member data?

Do the non-member data need to come from the same distribution as the member data? If so, how would one acquire this data in practical settings? Is it robust to pollution from the training set?

What if you do have access to the logits? How does this impact evidence gathering rate compared with probability estimation?

---

> ### Author Response · Authors · 2025-11-24
> **Responses to Reviewer roVQ (1/2)**
>
> Thank you for your positive, encouraging, and constructive feedback. We are happy that you consider our work both valuable and interesting for the ICLR community, and that you recommend it for acceptance.
> Below, we address each of your comments and questions individually:
>
> >**W1: The description of estimating token probabilities is not very clear. How is this semantic similarity measured? The “ground truth sequence” appears to be the one you would like to estimate the probability for – it would be helpful to refer to it as that if so.**
>
> We compute per-token embeddings using a sentence embedding encoder, such as sentence-transformers/all-MiniLM-L6-v2. For each next token, we calculate the similarity (using the dot product) between the embedding of the predicted next token from the target model and the embedding of the corresponding ground truth token. This similarity score is then converted into a probability using our sigmoid-based function, as specified in Equation 1.
>
> Our approach simplifies and accelerates previous methods, which relied on surrogate models to estimate token probabilities from the target model and incurred unnecessary computational overhead. We have updated the main text in the revised version of our submission (Section 3 on Estimating Token Probabilities) to reflect this approach. The code used for estimating token probabilities is available in the supplementary material (see: BADI/main/raw_values.py).
>
> >**Q1: How sensitive is the method to choice of non-member data?**
>
> To further demonstrate the robustness of our method to the selection of non-member data, we conducted an additional experiment. Specifically, we first shuffled and then partitioned the Ubuntu test dataset (used as non-member data) into subsets of 100 data points each. For every subset, our method consistently surpassed the 1% significance level, (e-values > 100). These results are presented in Figure 18 of Appendix D7. We extended this experiment to other subsets of the Pile, including PhilPapers, StackExchange, and YouTube Subtitles. Across all these cases, our method demonstrated robust performance.
>
> >**Q2: Do the non-member data need to come from the same distribution as the member data? If so, how would one acquire this data in practical settings? Is it robust to pollution from the training set?**
>
> Indeed, a key requirement of dataset inference is that non-member data should come from the same distribution as the member data, otherwise the risk of false positives increases. Several prior works have proposed different strategies for acquiring the held-out sets. For example, [1] suggests generating held-out sets by identifying natural identifiers, which are unique strings such as SSH keys created from random seeds in the suspect set, and then producing new identifiers from the exact same distribution to provide the held-out set. [2] demonstrates how to fully generate a held-out set, ensuring its distribution matches that of the suspect set with a post-hoc calibration. Finally, [3] proposes generating multiple rephrasings of the suspect set, each embedding a watermark with a unique secret key, then one rephrased version is released publicly, while the remaining versions are kept private to serve as held-out sets.
>
> We conducted an additional ablation study to evaluate the robustness of our method to potential “pollution” in the held-out set. We replace a specified proportion of non-members  with true training members. We then apply our hypothesis testing by betting method to the Ubuntu subset of the Pile dataset using the Pythia-12B model. In this framework, the bettor adaptively places stakes based on all observations seen so far.
>
> Under the alternative hypothesis, accumulated evidence causes the wealth process to grow. We execute the experiment over 50 random seeds and report the average wealth trajectory. We report the results in the new Figure 17 in the revised version of our submission. For pollution levels up to 70%, the test reliably gathers sufficient evidence to cross the 1% significance threshold within the first 200 observations. At 80% pollution, the rate of wealth growth diminishes, but the trajectory still shows consistent upward movement, indicating that the remaining clean examples provide enough evidence against the null. At 90% pollution, the curves flatten substantially, reflecting that the contaminated distribution becomes nearly indistinguishable from the null.
>
> Overall, the results demonstrate that the betting-based sequential test is robust to substantial pollution of the held-out set (validation data). Even with as much as 70% of the non-members replaced by members, the procedure continues to accumulate evidence and successfully rejects the null hypothesis.

---

> ### Author Response · Authors · 2025-11-24
> **Responses to Reviewer roVQ (2/2)**
>
> >**Q3: What if you do have access to the logits? How does this impact evidence gathering rate compared with probability estimation?**
>
> We ran additional experiments with the gray-box MIs instead of the black-box ones. We observe that the evidence is gathered faster, namely fewer data points are required to surpass the threshold of 1% (e-value >100). We show the new results in section 5.5. The overall growth of wealth for gray-box DI is faster than black-box  setting. However, as it is presented in Figure 4 of the paper, the gap between BADI and Gray-Box setting remains very small reflecting the effectiveness of our method in estimating token-probabilities.
>
> >**Writing: Line 131 consider rephrasing “However, most existing MI methods for LLMs, even the stronger gray-box attacks do not perform better than random guessing.” (...) Line 152: grammar issue “Consider a randomly ordered points” Line 160 “BADI rely” -> “BADI relies” Typos: Line 160 “BADI rely” -> “BADI relies”**
>
> We thank the Reviewer for carefully reading our paper, we really appreciate it. We corrected the typos in the updated version of our submission.
>
> **References:**
>
> [1] Rossi et al. *"Privacy Auditing for Large Language Models with Natural Identifiers"* ICLR 2025 Workshop Data Problems. https://openreview.net/forum?id=Ywdc1Vfoth#discussion
>
> [2] Zhao et al. *“Unlocking Post-hoc Dataset Inference with Synthetic Data”* ICML 2025 https://openreview.net/forum?id=a5Kgv47d2e
>
> [3] Rastogi et al. *“STAMP Your Content: Proving Dataset Membership via Watermarked Rephrasings”* ICML 2025
> https://openreview.net/forum?id=qF6mxani2X
>
> ---
>
> We hope that our responses and newly added experimental results satisfactorily address the Reviewer’s comments. We would appreciate a reconsideration of the evaluation should the clarifications and revisions resolve the issues raised.

---

### Author Response · Authors · 2025-11-24
**Rebuttal Summary**

We thank the Reviewers for their constructive feedback, which improved our work’s scope, rigor, and clarity. We appreciate the recognition that BADI addresses a challenging dataset inference problem under “flexible resource expenditure” (roVQ), that our token probability estimation is “novel” (ZAf3), and that the paper is “well-organized” (Xq4R, ck6T). In response, we revised the submission and added new experiments and analyses. A summary of updates follows.

**Our new and extended experiments:**

**Sensitivity Analysis for Non-Member Selection**: We provide an ablation study that evaluates the sensitivity of our betting-based framework to the choice of non-member data. (*Reviewer: roVQ*)

**Robustness to Contaminated Non-Member Sets**: We analyze the impact of non-member contamination on test validity and robustness. (*Reviewer: roVQ*)

**Real-World Black-Box Evaluation**: We apply our method to a commercial API (from OpenAI using GPT-3.5-Turbo) and on the BookMIA dataset to demonstrate the applicability of BADI applicability in real-world black-box scenarios. (Reviewer: Xq4R)

**Scalability to Newer and Larger Models**: We extend our evaluation to more recent and larger-scale models (e.g., Qwen2-7B, OLMo-7B) across different model families. (Reviewer: ck6T)

**Applicability to aligned models**: We also applied our method to the Qwen2-7B-Instruct model and compared it to the base model, where results indicate that our method is applicable for instruction tuned models. We also present the results for the GPT-3.5-Trubo model. (Reviewer: ZAf3)

**Betting Strategy Ablations and Hyperparameter Discussion**: We compare multiple betting strategies, assess their impact on test power, and provide guidance on selecting test hyperparameters. (*Reviewer: Xq4R*)

**Additional improvements:**

**Enhanced Comparison to Baselines**: We include ROC and AUC evaluations against PETAL and baseline methods, following suggestions from Reviewer: ck6T.

**Experimental Setup and Ablation Studies**: We expand the explanation of the experimental setup and explicitly justify the choice of the factor of 2 in Equation 1, as requested by *Reviewer: Xq4R*.

**Main Comments:**

> **BADI Scales Across Model Sizes and Families**

Addressing Reviewer ck6T, we expanded our experiments beyond Pythia to a thorough evaluation on OLMo-7B (2.5T tokens). Appendix E shows that BADI consistently surpasses baseline methods, with full results in Table 10. We also tested BADI on additional models, including Qwen2-7B (7T tokens) and GPT-3.5-Turbo, which demonstrate successful membership detection (Figs. 20–21).

> **BADI Works on Real Commercial Black-Box APIs**

In response to Reviewer: Xq4R, we evaluated BADI on OpenAI’s GPT-3.5-Turbo using the BookMIA dataset. Appendix E shows that BADI detects the effect with fewer than 200 queries.

> **Testing-by-Betting Resists Non-Member Selection and Pollution**

Following Reviewer: roVQ, we ran two ablations:
1. **Robustness to Non-Member Selection:**
   Using the Ubuntu subset with Pythia-12B, we tested DI on random batches (100 samples each). Every batch consistently surpasses the 1% threshold (Fig. 18, Appendix D7).
2. **Robustness to Contaminated Non-Members:**
   Varying contamination levels show testing remains effective until ~70% contamination, where overlap reduces evidence (Fig. 17, Appendix D6), demonstrating robustness in noisy or moderately adversarial settings.

> **Gray-Box DI Requires Fewer Observations**

 We compared evidence accumulation using actual token probabilities (gray-box) versus estimated ones (black-box). Gray-box testing reaches the 1% threshold faster (Fig. 4), though BADI still efficiently accumulates signal in black-box settings. (Reviewer: roVQ)

> **Testing-by-Betting is Robust to Strategies and Hyperparameters**

Addressing Reviewer: Xq4R, Appendix D5 provides test power analysis, and Appendix C1 adds Approximate Kelly Betting.
1. Table 9 shows consistent results under Kelly Betting.
2. Fig. 16 shows stable power for both strategies under varying caps.
Betting values should remain within (-1, 1), consistent with log-optimality.

**Additional notes:**

1. **Empirical justification for factor 2 in Eq. 1:**
   Appendix A.6 (Fig. 6) evaluates scaling factors {0.5, 2, 10}. Factor 2 provides the strongest separation between members and non-members.

2. **Improved comparisons with PETAL and baseline:**
   Following Reviewer: ck6T, we now include ROC curves and AUC comparisons (Fig. 14). Numerical AUC values across Pile subsets are reported in Tables 1–3, showing superior performance.

3. **Improved log-probability estimation compared to baseline and Petal**
Figure 3 shows that our method outperforms PETAL and Baseline, reaching the 1% significance threshold with fewer observations. A comparison in Figure 4, where the gap between BADI and the gray-box setting is small, confirms our method's success in separating log-probabilities and detecting membership.

---

### Meta-Review · Area_Chair_itzd · 2025-12-03

**Summary:**

This paper proposes BADI, a black-box and anytime-valid framework for dataset inference in LLMs, aiming to identify if a specific dataset was used in training. The method estimates token probabilities from label-only API outputs and uses e-values with sequential testing for flexible, statistically valid auditing. Reviewers recognized the practical motivation and novel aspects, such as the token probability estimation. Initial scores were mixed: one reviewer gave a 6 ("marginally above acceptance"), while three others gave a 4 ("marginally below acceptance"). Key concerns centered on limited empirical validation. Reviewers noted the experiments relied heavily on the older Pythia model family and lacked testing on modern, large-scale LLMs trained on trillions of tokens and commercial black-box APIs. Other concerns involved the robustness of non-member data selection, the justification for specific methodological choices (e.g., the sigmoid scaling factor), and insufficient comparison to baseline methods. In the rebuttal, the authors conducted new experiments, applying BADI to GPT-3.5-Turbo, Qwen2-7B, and OLMo-7B models, and added analyses on robustness and betting strategies. While these additions partially addressed some concerns, the core issue regarding the method's proven effectiveness on contemporary, web-scale LLMs remains inadequately resolved for the standards of this conference. I don't think 7B-size models and models such as Qwen2 and GPT-3.5 are big enough and recent enough to demonstrate the scaling ability and the utilization of the proposed method.

**Reviewer Concerns:**

The rebuttal addressed specific technical questions about non-member data robustness, hyperparameter choice, and added experiments on more models. However, the outstanding major concern is that the method's evaluation is still not sufficiently convincing for state-of-the-art, massively scaled LLMs, which is critical for the paper's core claim of practical applicability.

**Reviewer Scores:**

Reviewer roVQ's score (6) might have remained unchanged given their positive initial assessment. Reviewer ZAf3 and Xq4R, who raised significant concerns about applicability and empirical validation, likely would not have raised their scores (4) significantly, as the new experiments did not fully alleviate doubts about modern LLM scalability.

---

### Decision · Program_Chairs · 2026-01-26

Reject